# MCDM-Based R&D Project Selection: A Systematic Literature Review

**Dalton Garcia Borges de Souza** [1,2,3,*] ⓘ**, Erivelton Antonio dos Santos** [4,5] ⓘ**, Nei Yoshihiro Soma** [1,2,6]
**and Carlos Eduardo Sanches da Silva** [4,7]

1   Division of Computer Science, Aeronautics Institute of Technology, São José dos Campos 12.228-900, Brazil;
    nys@ita.br
2   Institute of Science and Technology, Federal University of Sao Paulo, São José dos Campos 12.247-014, Brazil
3   Lorena School of Engineering, University of São Paulo, Lorena 12.602-810, Brazil
4   Institute of Industrial Engineering and Management, Federal University of Itajubá, Itajubá 37.500-903, Brazil;
    erivelton.santos@unifenas.br (E.A.d.S.); sanches@unifei.edu.br (C.E.S.d.S.)
5   Department of Administration Course, José do Rosário Vellano University, Alfenas 37.132-440, Brazil
6   Department of Mechanical Engineering, University of Taubate, Taubaté 12.020-270, Brazil
7   Secretariat of Higher Education, Ministry of Education, Brasília 70.047-900, Brazil
*   Correspondence: dalton.borges@unifesp.br

**Abstract:** From small spin-offs deploying innovative software to big pharmaceutical complexes making vaccines, Research and Development (R&D) Project Portfolio Selection (PPS) is an essential strategic process for various companies. It was never easy to select a set of projects among many feasible possibilities, even for yesterday's paces. However, the world is rapidly changing, and so is R&D PPS. The portfolio objectives excel profit in the same manner that model constraints go beyond budget limitations. In parallel, project selection approaches and solving algorithms followed the increase of computational power. Despite all those changes, the importance of Multi-Criteria Decision Making (MCDM) methods and the decision criteria used for R&D PPS, there is still room for a systematic literature review (SLR) for the topic. Thus, this paper offers an SLR of the existing literature from the half-century, 1970, and onward MCDM-based R&D PPS performed in Scopus and Web of Science Core Collection. We provide a comprehensive picture of this field, show how it is changing, and highlight standard practices and research opportunities in the area. We perform a broad classification of the MCDM methods, categorized by the nature of alternatives, types of integration approach, the MCDM method itself, and types of uncertainty, by the 66 studies in the SLR database. The portfolios' classification obeys the application domain and the number of projects. We have also explored all the 263 criteria found in the literature by grouping them according to experts from five Brazilian R&D organizations that together manage portfolios valued around US$ 5 billion a year, accounting for 38% of all Brazilian annual expenditure in R&D projects. We also include a bibliometric analysis of the considered papers and research opportunities highlighted or not explored by researchers. Given the increasing number of decision-making approaches and new technologies available, we hope to provide guidance on the topic and promote knowledge production and growth concerning the usage of MCDM methods and decision criteria in R&D PPS.

**Keywords:** research and development; portfolio management; decision support systems; multi-attribute decision making; systematic literature review

## 1. Introduction

In the last 50 years, many authors have suggested project selection approaches for different topics such as healthcare, construction, and the public sector. One of the most relevant topics is R&D, with many relevant articles indexed in scientific databases. Thore [1] suggests that, by the end of the 20th century, as a consequence of the unbridled increase of communication and information technology, a new economy has arisen, recognizing it as

the Knowledge economy. The drivers of this latest economy are the R&D projects. That is why R&D managers constantly need to develop systems and procedures, which will improve the likelihood of success of their business. According to UNESCO Institute for Statistics (UIS), the yearly global spending on R&D projects reached a record of almost US$ 1.7 trillion in 2018 [2]. While poor and middle-income countries invest in R&D less than developed countries, the numbers are bettering in some cases [3]. During the COVID-19 pandemic, R&D investment may increase even more, at least for the largest R&D spenders, who tend to increase R&D investment during financially struggling times, such as the worldwide financial crisis in 2008 [4].

In Brazil, for example, organizations have invested increasing amounts of money in R&D over the last two decades, despite the negative gradient in the last two years. The US$ 21 billion they disbursed in 2016 is almost six times the amount of that in 2000. Slightly more than half of these values come from public sources, which makes the Brazilian government the main engine for R&D and innovation in the country [5]. This proportion is quite different in developed countries such as Germany, Japan, and the United States, where during standard times less than 30% of the total invested in R&D comes from the public purse [6].

Regardless of who controls the company, they will consolidate R&D investments mainly by implementing projects, whose selection occurs according to their alignment to the organization's goals [7]. However, the associated risks in performing R&D projects have proved to have a significant impact since the choice of unsuitable projects may result in a consequential loss of financial and human resources [8,9]. In this case, the impact of corporate strategy is usually perceived correctly in the selection of R&D projects [10]. Thus, joining all projects with the organization's strategic direction is crucial to utilizing the resources better [11–13]. We know this practice as R&D Project Portfolio Selection (PPS), a branch of the Project Portfolio Management (PPM) field of study. R&D PPS is commonly a multicriteria process, with model and criteria selection processes as two of its main steps [14].

The decision-making process in R&D PPS is similar to decision-making in other domains. Therefore, the decision-making frameworks do not conceptually change depending on the portfolios' characteristics and application domain. However, the used selection methods have changed, and many scientific papers address diverse methodologies for R&D PPS. The main differences and challenges in R&D PPS are (a) the expenditure in projects expresses sizable investments; (b) the enterprises make those investments in their future; thus, (c) the projects need be tied to the corporate strategy; and (d) the R&D projects returns have extended lead times, are risky and multidimensional; (e) the environment is rough, and the results changeable [15–17]. These unique characteristics make it challenging to perform suitable or optimal decisions.

On the other hand, R&D PPS still have difficulties that are shared by PPS in other fields. Commonly, the selection process may consider:

1. A big portfolio, with several projects [18–20];
2. Qualitative and quantitative data [21,22];
3. Uncertainty generated by imprecise information [23–25];
4. Uncertainty generated by limited data [26–28];
5. Multiple interdependent and/or conflicting criteria (attributes and/or goals) [29];
6. Interdependence and interrelation among projects [30–32];
7. Mutually exclusive projects or cannibalization [9,33–36];
8. Resource constraints [37,38];
9. The optimal schedule [39–41];
10. Human resource allocation [42];

MCDM methods are recurrent on R&D PPS by assisting the decision-makers in ranking and choosing the most suitable alternative based on several conflicting criteria. Regarding the methods used in PPS, MCDM methods are the highly scientifically investigated approaches. MCDM holds the decision-makers in ranking or electing the best alternatives

based on numerous, sometimes conflicting, criteria. They cover from plain [43] to elaborate approaches [17], from typical [44] to singular [45], and from single [16] to combined ones [46].

The importance of the R&D PPS subject, to the best of our knowledge the literature gap remains in contributing a broad picture of the role of MCDM methods in R&D PPS through classifying, comparing, and analyzing the various MCDM approaches used. We aim to explore the area and provide a state-of-the-art reference of MCDM-based R&D PPS. This paper aims to systematically collect and analyze papers published on the subject from 1970 to 2020 and are available through two best-known databases available: Scopus and Web of Science Core Collection. Recently, similar approaches have been used in other fields of study [47,48].

The main innovations of this paper are in the extension of the explorations and analysis performed, and on the perspectives put on the R&D PPS field of study. We also position our investigation in the light of other up-to-date studies. For instance, Afshari [49] highlights the importance of the criteria used during project selection processes and advocates for the creation of systematic frameworks that may help decision makers on finding the most suitable criteria. Almeida et al. [50] indicate the need for scientific material that assist researchers on selecting MCDM methods for a given application. Souza et al. [51] discuss the importance of knowing the R&D PPS ambience to facilitate finding similar approaches that have already been developed. Considering this, the research questions addressed in this review are:

- RQ1: Methods. Which MCDM methods are used in R&D PPS? Which is the nature of their alternatives? Are the methods used as individual or integrated approaches? What are the most frequently used MADM (Multi-attribute decision making) and MODM (Multi-objective decision making) methods? How has the usage of those methods changed with time? How are they used? How do they consider uncertainty?
- RQ2: Portfolios. How big are they? Which application domains are the most explored? Which software, solvers or programming languages are employed in the selection process? How much attention do the papers give to the criteria used?
- RQ3: Research Field. Can the publication timeline be split into periods of theory intensification? Which are the most cited articles? Who are the top authors? Where are they from?
- RQ4: Whole data. Which data are correlated? Which conclusions can be made by looking at those correlations?
- RQ5: Criteria used. Which criteria are used by the authors? Are all those criteria expressing different perspectives? Or could they be summarized into a smaller list of criteria?
- RQ6: Research opportunities and trends. Which extensions of previous works could be done? Which research opportunities could be explored?

In this work, we attempt to answer those questions through a full extent mapping of approach's natures, integration's approach, papers correlations analysis, software's types, portfolio's size, uncertainty models, list and explain the criteria and bibliometric analysis. Hopefully, these findings of our analysis will be beneficial to the community of academics and practitioners in R&D PPS.

The remainder of this paper is arranged as follows: Section 2 introduces the methodology and presents the foundations underlying the next three sections and systematically reviews the literature on MCDM-based R&D PPS. We present a timeline of the utilization of the main methods and perform a broad classification of all MCDM methods used in the last 50 years. We categorized them according to the nature of alternatives (MADM—Multi-attribute decision making—and MODM—Multi-objective decision making), type of integration approach (Individual and Integrated), the MCDM method itself and types of uncertainty (Deterministic, Probabilistic and Fuzzy). The portfolios' classification follows the application domain and number of projects. In Section 3, we present the criteria used by the SLR papers. We also propose a smaller list of criteria that, at least for the main five

Brazilian R&D public organizations, represent the perspectives which are explored by the 263 criteria proposed in the literature. We present a bibliometric analysis of the considered papers in Section 4, which provides trends on the topic and information about papers, authors, and their countries. A short Section 5 highlights opportunities in MCDM-based R&D PPS. Finally, Section 6 presents the conclusions.

## 2. Methodology

Our paper reports a comprehensive literature review about MCDM-based R&D PPS. The research methodology follows the recommendations of Rowley and Slack [52] on how to conduct a systematic literature review, and the PRISMA framework [53] (Preferred Reporting Items for Systematic Reviews and Meta-analyses). We also based our work on the frameworks proposed by Jahangirian et al. [54], Diaby et al. [55]. In addition, we adopted and expanded it to pre-search steps, as shown in Figure 1 and the Figure A1 in the Appendix. The purpose of our work is also similar to the ones conducted by Harrison et al. [47] and Pourhabibi et al. [48] that recently proposed an SLR of portfolio optimization for defense applications. Nevertheless, we addressed it to a different field of study, and we applied different researches tools, such as the online application Parsifal® as a checklist to guide the SLR process, and we also used the Prisma checklist and the Prisma abstract checklist, as shown in the supplementary materials. The PRISMA statement intends to assist authors in improving the reporting of systematic literature reviews through a 27-item checklist [53].

Firstly, we perform an exploratory search to obtain the most cited articles on MCDM bibliometric analysis and literature reviews [56–66]. We got from these articles many domains and several keywords associated with the MCDM searches. These keywords are acronyms, synonyms, and equivalents words to MCDM and its highly cited methods. Afterward, we combined them with keywords related to R&D and PPS. Thus, we used a total of 134 keywords that gave a total of 2604 Boolean combinations to find articles related to MCDM-based approaches in R&D PPS. The articles were found according to their title, abstract, and article keywords. The articles are filtered according to Table 1. The searching keywords, which were taken from [67], can be seen in Table 2.

**Table 1.** Inclusion and exclusion criteria.

| Step | Inclusion Criteria | Exclusion Criteria |
|---|---|---|
| Title screening | Articles related or that could be related to project portfolio in general. Since the title briefly introduce the main topic of the article, many works that performs MCDM-based R&D PPS as a secondary topic could fall out the SLR. | Papers that do not present approaches or cases related to project portfolio were left out. This is the case of articles introducing MCDM methods to general applications, other fields of study or other subjects inside the big area of project management, such as expert assessment, market assessment and performance evaluation of already concluded projects. |
| Abstract screening | Articles related to R&D PPS in general. Since some abstracts do not present the methods and approaches employed, we have decided to check this information later. | Additionally to the exclusion criteria performed in the first step, were left out the SLR articles that are addressing PPS to other areas rather than R&D, or are selecting other elements rather than projects, such as technology, suppliers, products and others. |
| Text screening | Only articles that present MCDM-based approaches to select R&D projects. | Additionally to the exclusion criteria performed in the two previous steps, articles addressing mono-criteria project selection or that do not find a set of optimal or recommended projects were left out. |

**Table 2.** Strings used to perform the search.

| | **Boolean Combination: (MCDM OR MADM OR MODM OR Methods) AND PPS AND R&D** |
|---|---|
| MCDM | "MCDM" OR "multicriteria decision making" OR "multi-criteria decision making" OR "multi criteria decision making" OR "multiplecriteria decision making" OR "multiplecriteria decision making" OR "multiple criteria decision making" OR "MCDA" OR "multicriteria decision analysis" OR "multi-criteria decision analysis" OR "multi criteria decision analysis" OR "multiplecriteria decision analysis" OR "multiple-criteria decision analysis" OR "multiple criteria decision analysis" OR "multicriteria decision aiding" OR "multi-criteria decision aiding" OR "multi criteria decision aiding" OR "multiplecriteria decision aiding" OR "multiple-criteria decision aiding" OR "multiple criteria decision aiding" |
| MADM | "MADM" OR "multiattribute decision making" OR "multi-attribute decision making" OR "multi attribute decision making" OR "multipleattribute decision making" OR "multiple-attribute decision making" OR "multiple attribute decision making" OR "MADA" OR "multiattribute decision analysis" OR "multi-attribute decision analysis" OR "multi attribute decision analysis" OR "multipleattribute decision analysis" OR "multiple-attribute decision analysis" OR "multiple attribute decision analysis" OR "multiattribute decision aiding" OR "multi-attribute decision aiding" OR "multi attribute decision aiding" OR "multipleattribute decision aiding" OR "multiple-attribute decision aiding" OR "multiple attribute decision aiding" |
| MODM | "MODM" OR "multiobjective decision making" OR "multi-objective decision making" OR "multi objective decision making" OR "multipleobjective decision making" OR "multiple-objective decision making" OR "multiple objective decision making" OR "MODA" OR "multiobjective decision analysis" OR "multi-objective decision analysis" OR "multi objective decision analysis" OR "multipleobjective decision analysis" OR "multiple-objective decision analysis" OR "multiple objective decision analysis" OR "multiobjective decision aiding" OR "multi-objective decision aiding" OR "multi objective decision aiding" OR "multipleobjective decision aiding" OR "multiple-objective decision aiding" OR "multiple objective decision aiding" |
| Methods | "Simple Additive Weighting" OR "Additive Ration Assessment" OR "SWARA" OR "Step-wiseWeight Assessment Ration Analysis" OR "TOPSIS" OR "Technique for Order of Preference by Similarity to Ideal Solution" OR "ELECTRE" OR "Elimination et Choix Traduisant la Réalité" OR "Elimination and Choice Expressing REality" OR "LINMAP" OR "Linear Programming Technique for Multidimensional Analysis and Preference" OR "AHP" OR "Analytic Hierarchy Process" OR "ANP" OR "Analytic Network Process" OR "PROMETHEE" OR "The Preference Ranking Organization Method for Enrichment of Evaluations" OR "MOORA" OR "Multi-Objective Optimization on the basis of Ration Analysis" OR "MULTIMOORA" OR "Multiplicative form with Multi-Objective Optimization on the basis of Ration Analysis" OR "DEA" OR "Data Envelopment Analysis" OR "VIKOR" OR "Visekriterijumska optimizacija i Kompromisno Resenje" OR "Multicriteria Optimization and Compromise Solution" OR "COPRAS" OR "Complex Proportional Assessment" OR "EVAMIX" OR "Evaluation of Mixed Data" OR "DEMATEL" OR "Decision-Making trial and Evaluation Laboratory" OR "WASPAS" OR "Weighted Aggregated Sum Product Assessment" OR "WSM" OR "Weighted Sum Method" OR "WPM" OR "Weighted Product Method" OR "Compromise Programming" OR "MAUT" OR "Multi-Attribute Utility Theory" OR "CBR" OR "Case Based Reasoning" OR "Genetic Algorithm" OR "SMART" OR "Simple Multi-Attribute Rating Technique" OR "MAVT" OR "Multi-Attribute Value Theory" OR "REMBRANDT" OR "Ratio Estimation in Magnitudes" OR "Decibels to Rate Alternatives which are Non-Dominated" OR "NAIADE" OR "Novel Approach to Imprecise Assessment and Decision Environments" OR "Linear Programming" OR "Non-Linear Programming" OR "Non Linear Programming" OR "Multi-Objective Programming" OR "Multi Objective Programming" OR "Multiobjective programming" OR "Goal Programming" OR "Integer Linear Programming" OR "Integer Non-Linear Programming" OR "Integer Non Linear Programming" OR "Integer Programming" |
| PPS | "Project Selection" OR "Project Evaluation" OR "Project Portfolio Selection" OR "Project Portfolio Evaluation" OR "Project Portfolio" OR "Project Portfolio Management" |
| R&D | "Research and Development" OR "Research & Development" OR "R&D" OR "R and D" OR "RnD" OR "R n D" OR "R & D" |

Source: Exactly as proposed by [67].

We performed this search in the two main widespread databases available: Scopus®, the largest multidisciplinary database, including approximately 15,000 peer-reviewed journals and over 4000 publishers [54]; and Web of Science® Core Collection, a database that includes around 10,000 peer-reviewed journals and it was for years the only citation database covering all scientific research domains [68]. However, more articles can be found outside of those databases; the scope of this paper was restricted to only papers available in those two. The search started in 1 January 2019, and the last update finished in

26 January 2021. A total of 314 results from 1970 (the year when the first article is dated) to 2020 could be found. From those, we have considered only non-duplicated articles in English and peer-reviewed published journals for the next steps. Then, we performed three screening steps. Firstly, the titles were analyzed, and we rejected articles that did not suit the scope of this work. Afterward, articles were rejected based on their abstracts and subsequently on the full text. A total of 66 articles were finally selected (see Appendix on Tables A1 and A2). Table 1 shows the inclusion and exclusion criteria as well as the corresponding screening steps. Every step also considers the inclusion and exclusion criteria of subsequent steps. For pattern, authors 1 and 2 independently reviewed all the studies, then reached a final database consensus, as shown in Figure 1. Moreover, as an example of the selection process, although the studies of Huang and Chu [69], and Souza et al. [51] might appear to match the inclusion criteria, we excluded them because they did not focus on R&D subject.

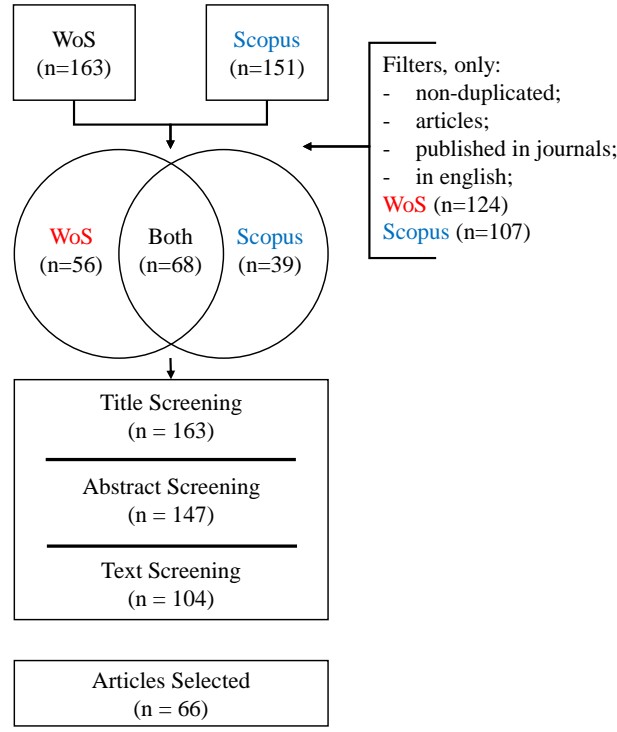

**Figure 1.** The filters applied.

MCDM approaches classification variation as the classification criteria. Concerning the nature of the alternatives, they are Multi-Objective Decision-Making (MODM) and Multi-Attribute Decision-Making (MADM), or an aggregate of both (see Figure 2). MODM methods have no predetermined options, and the optimal option selection depends on an infinite and continuous number of circumstances subjected to a set of constraints. Generally, MODM methods include mathematical approaches, i.e., integer linear programming, goal programming, linear programming, integer non-linear programming, multi-objective programming [58]. On the other hand, MADM methods deal with a discrete and finite number of options designated by a predetermined set of criteria. Thus their main task is to achieve a reasonable selection, assessment, and grading among the viable possibilities [66]. AHP (Analytical Hierarchy Process) and TOPSIS (Technique for Order Preference by Similarity to Ideal Solution) are relevant examples of MADM methods. Therefore, some authors refer to MODM and MADM problems as continuous and discrete problems, respectively [59]. Notwithstanding, PPS problems use knapsack problems solutions since the latter involves just discrete input data for each project. Thus, MODM methods are commonly constrained to work with discrete alternatives. Therefore, the classification in

MADM and MODM considers the widespread application of the method. In this work, we follow the classification made by Chai et al. [70], that includes DEA as an MODM method.

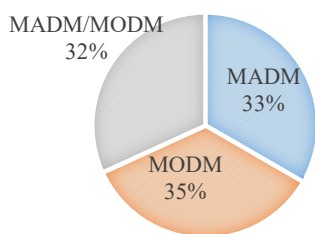

**Figure 2.** Nature of the alternatives.

According to their methodology, MCDM methods classification follows the individual methodology approach and integrated methodology approach, which depend on the number of methods integrated into them (see Figure 3) [57]. The work of Meadeand Presley [16] is a pertinent example of an individual methodology approach using ANP (Analytical Network Process), a MADM method. On the other hand, Bard et al. [44] introduce 0–1 integer programming as an individual MODM approach method to select the optimal project portfolio. Liberatore [15] shows how to integrate MADM and MODM methods by coupling AHP and 0–1 integer linear programming into an integrated approach. The proportion between the type of integration approach also seems to be constant over all periods analyzed. However, the integrated methods have changed: today, we increasingly integrate two or more different MADM methods (MADM-MADM integration). Indeed, until 1995 there were only MADM-MODM and MODM-MODM integrations. Another possible analysis is that papers addressing more than one model or comparing models do not appear too often as in the past. For instance, the last paper addressing both individual and integrated approaches dates 1988. It also reflects on the greater acceptance for specific articles today, rather than the generalist ones.

It is worth mentioning that the usage of MODM methods as Individual Approaches and Integrated Approaches presents a moderate positive Pearson correlation coefficient (0.54) and a moderate negative correlation coefficient (−0.52), respectively. Generally, those articles employ only a few criteria, which are used as objective functions or constraints to the problem. MADM methods have a leading role in integrated approaches: criteria weight. In this case, a considerable number of papers do not even explain the criteria used, which is also pointed out by the moderate negative correlation (−0.51) between Linear Programming (So far, the most frequently used MODM method) and the presence of explained criteria on the paper. The information presented by Figure 3 is desegregated by year in Figure 4. It makes it visually easy to observe the increasing in use of integrated approaches in the last two decades, specially during the last five years.

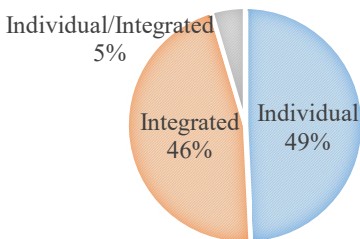

**Figure 3.** Type of integration approach.

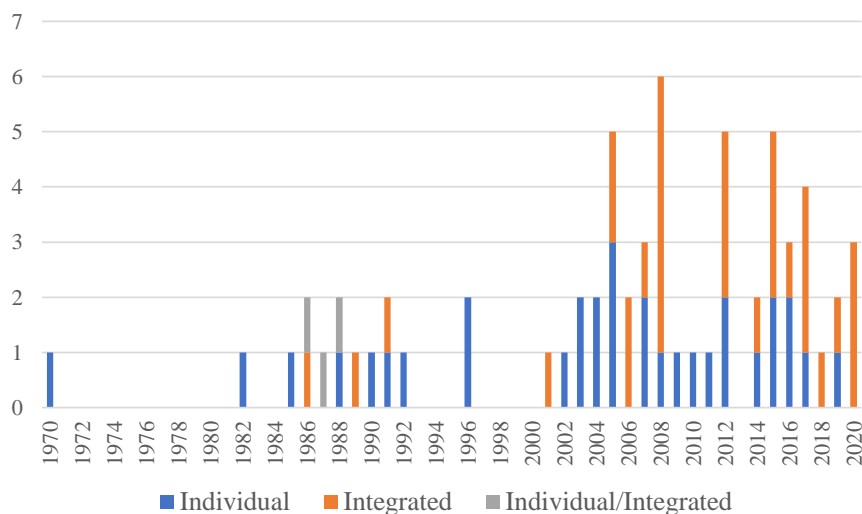

**Figure 4.** Counting the integration approaches per year.

Observe that the pattern regarding the nature of alternatives changes over the years, Figure 5. From 1970 to 1995, the only methods used were MODM and MADM/MODM. That period coincides with the publication of the first PMBoK (Project Management Body of Knowledge), in 1996 [71], and will define here the first period of theory intensification. There appeared many forms of 0–1 integer programming in several articles [13,15,31,42,44,72]. In the same period, AHP was the most integrated MADM method, exclusively with 0–1 integer linear programming and by Liberatore [12,13,15]. From 1996 to the present, the second period of theory intensification saw the emergence of individual MADM, and integrated MADM-MADM approaches. AHP and its variations were the most frequently used individual methods [73–76], followed by ANP [16,29] and ROA (Real Option Analysis) [22,23,25]. Regarding Integrated MADM-MADM approaches, there is a variety of combinations with commonly used methods, such as AHP and DEA [10,43]; TOP-SIS [21,77,78]; and DEMATEL (Decision Making Trial and Evaluation Laboratory) [46,79,80]. Figures 6 and 7 introduce the most frequently used MADM and MODM methods, respectively. Notice that Figure 6 shows only the most frequently used methods; other methods correspond for 14.9% of the total. The meaning of the main MADM methods acronyms are in Table 3, as well as their first reference on literature.

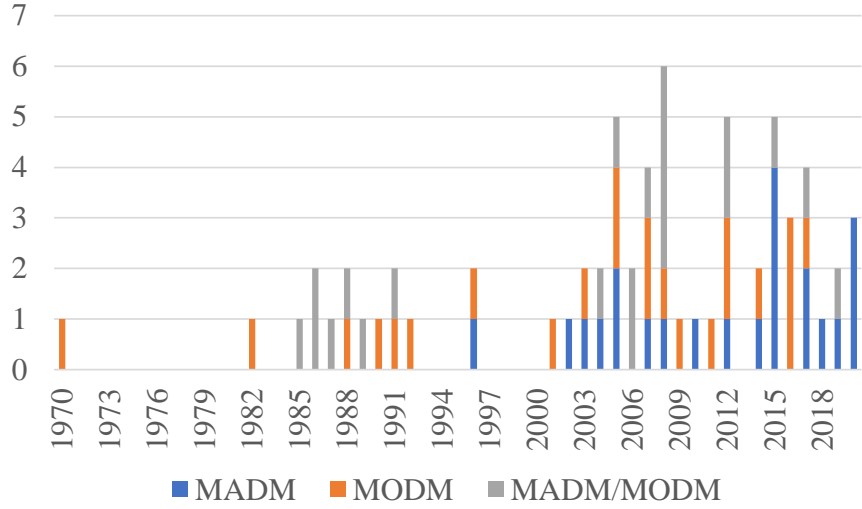

**Figure 5.** Number of publications by nature of alternatives over the years 1970–2019.

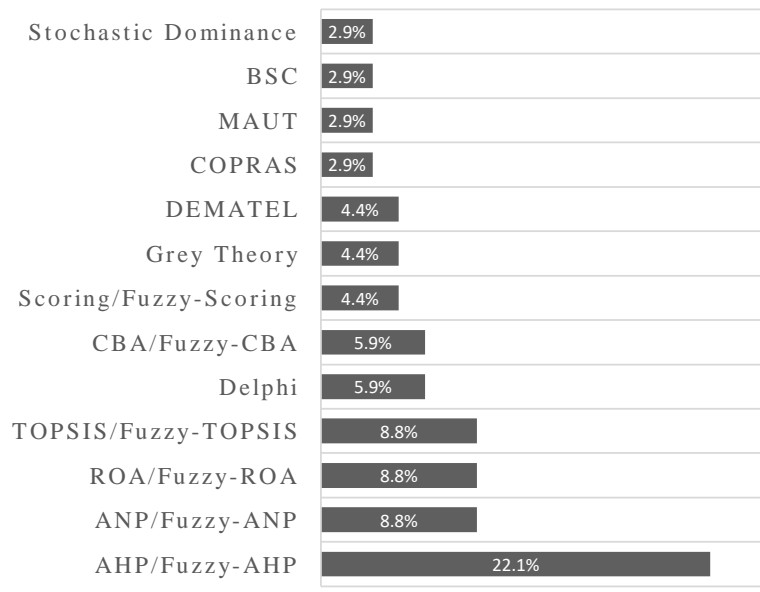

**Figure 6.** Most used MADM methods used in the articles.

**Table 3.** MADM methods acronyms, meanings and first references.

| Acronym | Method | First Reference | Year |
|---|---|---|---|
| AHP | Analytic Hierarchy Process | Saaty [81] | 1980 |
| ANP | Analytic Network Process | Saaty [82] | 2001 |
| BCG Matrix | Boston Consulting Group Matrix | Boston Consulting Group [83] | 1970 |
| BSC | Balanced Scorecard | Norton and Kaplan [84] | 1999 |
| CBA | Cost-Benefit Analysis | Mishan and Eauston [85] | 1976 |
| COPRAS | Complex Proportional Assessment | Zavadskas et al. [86] | 1994 |
| DEMATEL | Decision-Making trial and Evaluation Laboratory | Gabus and Fontela [87] | 1973 |
| ELECTRE | French: Elimination et Choix Traduisant la Réalité (Elimination and Choice Expressing Reality) | Benayoun amd Sussman [88] | 1966 |
| MAUT | Multi-Attribute Utility Theory | Keeney and Raiffa [89] | 1976 |
| PROMETHEE | Preference Ranking Organization Method for Enrichment of Evaluations | Brans and Vincke [90] | 1985 |
| ROA | Real Options Analysis | Trigeorgis [91] | 1995 |
| TOPSIS | Technique for Order of Preference by Similarity to Ideal Solution | Hwang [92] | 1981 |
| VIKOR | Serbian: Visekriterijumska optimizacija i Kompromisno Resenje (Multi-criteria Optimization and Compromise Solution) | Opricovic [93] | 2002 |

Among all MCDM methods for R&D PPS, AHP is the most relevant one, appearing in 15 papers. From those papers, 14 have their scopes presented and properly explained by Souza et al. [51], which presented an integration of Fuzzy-AHP Extent Analysis and Fuzzy-DEMATEL to criteria selection in R&D. The only paper not covered by Souza et al. is the one from Samanlioglu et al. [94], that presented an integration of hesitant F-AHP and hesitant F-VIKOR, in order to evaluate and rank innovation projects. In this case, hesitant F-AHP is applied to get fuzzy evaluation criteria weights, and hesitant F-VIKOR is applied to rank innovation project options. For completeness, we will not reproduce here the timeline of AHP in R&D Project Selection. We will focus on the other methods that are not explained by Souza et al. [51].

Another MCDM method similar do AHP is ANP, which appears in 8.8% of the articles. Mohanty et al. [95] show a utilization of fuzzy ANP (analytic network process) accompanying with fuzzy cost examination in selecting R&D projects, aiming to overwhelm the uncertainty in the preferences. The approach is interactive and built on two sets of critical factors. Initially, they screened projects to check if they were acceptable and

reasonably progressing toward completion. Then, the projects which failed the test are finished, and the projects remaining are weighed with candidate projects to determine which one should be included in the portfolio. Meade and Presley [16] discussed the use of the Analytic Network Process (ANP) and presented a generic model based on many factors and criteria to support different situations. Jung and Seo [29] explored the analytic network process (ANP) approach applied for the evaluation of R&D projects that are components of programs with complex objectives. Jeng and Huang [79] proposed a decision model for assessing a project portfolio at the beginning initiation stage, including a modified Delphi method (MDM), decision-making trial and evaluation laboratory (DEMATEL) method, and analytic network process (ANP). Mohaghar et al. [21] presented an integrated fuzzy approach, with Fuzzy-ANP and Fuzzy-TOPSIS, for selecting R&D projects. DEMATEL and ANP are moderately correlated in the articles (+0.55). It is justified by using the influence matrix given by DEMATEL as an input to ANP or Fuzzy-ANP.

As frequently as ANP, ROA (Real Option Analysis) is also well-used to select R&D Project Portfolios, appearing in 8.8% of the papers. Its first usage in the topic dates 2006 [25], remaining used until 2014 [23]. It is mainly used as a side method in integrated approaches along with other MADM or MODM Methods. It is interesting to mention that all applications of this method are given in the fuzzy environment, with the most realistic option valuation given by a fuzzy pay-off method [24,96,97].

TOPSIS/Fuzzy-TOPSIS appears five times and DELPHI method and CBA/Fuzzy-CBA (Cost-Benefit Analysis) were used four times each, followed by DEMATEL, Scoring/Fuzzy-Scoring methods and Grey Theory (three times each) and then MAUT (Multi-Attribute Utility Theory), BSC (Balanced Scorecard) and Stochastic Dominance (two times each). Other methods with one apparition each sum up for 14,9% of the MADM applications. In the case of CBA, it mainly appears as an auxiliary method in Integrated Approaches and shows a strong positive correlation (+0.86) with articles that introduce more than one approach to select project portfolios. A similar side role is performed by BCG (Boston Consulting Group) Matrix, BSC, and Scoring methods.

We can classify the MCDM methods in many ways [50]. A traditional classification divides them into unique criteria of synthesis methods, outranking methods, and interactive methods. We may also classify it into compensatory or non-compensatory methods. In this case, the preference relation will be compensatory if there are trade-offs among criteria and non-compensatory otherwise. Generally, the unique criterion of synthesis methods is also compensatory.

On the other hand, outranking methods use non compensatory rationality. Interactive methods cover the whole spectrum of MODM methods. If we look only at MADM methods, the most classic ones can fit into four categories [98].

- Multi-criteria value functions. The methods are commonly based on a value function, obtained by weighted summation or weighted multiplication. The criteria weights are non-negative and sum to 1. If weighted multiplication is used then criteria will be non compensatory, where a zero score on any individual value will result in an overall zero performance score.
- Outranking approaches. Those methods generally involve the identification of every pair of decision options $i$ and $i'$ giving $n^2 - n$ pairs in total. Outranking approaches also apply some utility function, containing criteria weights.
- Distance to ideal point methods. These methods calculate ideal and anti-ideal values for the criteria. Then, decision options that are closest to an ideal solution are preferred, while decision options closest to the anti-ideal solutions are avoided. The concept of Euclidean distance are normally adopted.
- Pairwise comparison methods. These methods compare each unique pair's criteria and alternatives, giving $n(n - 1)/2$ comparisons. The comparisons are made between criteria and also between decision options.

We compare those categories in Table 4, with advantages and disadvantages [99]. Notice that Table 4 lists only the methods used by the articles considered in this literature

review. Thus, well known outranking methods, such as ELECTRE (in French: ELimination Et Choix Traduisant la REalité—in English: Elimination and Choice Expressing Reality) and PROMETHEE (Preference Ranking Organization Method for Enriched Evaluation) families do not appear in R&D PPS context, which may be interesting in some occasions. Other not used and well know Distance to ideal point method is CP (Compromise Programming).

It is worth mentioning that only a few articles give proper explanations of why they have chosen a specific MCDM method in their R&D PPS context [26,100]. In fact, to the best of our experience, there is no framework available in the literature that helps researchers select the best methods in each PPS case and other MCDM applications [50].

The classification used in Figure 7 is the same used by Chai et al. [70]. Linear programming is the most frequently used MODM method with 17 appearances. Subsequently, 10 of them correspond to integer approaches and four to mixed-integer approaches. Multi-objective programming is the second most frequently used MODM method, appearing in nine papers. From those, seven are integer approaches, and only one has used non-linear data. DEA appears in five papers, followed by Non-Linear programming, used in four papers, with two integer approaches and one mixed-integer approach. Stochastic and Goal Programming appeared twice each. Goal programming, Linear and Non-Linear models appeared together.

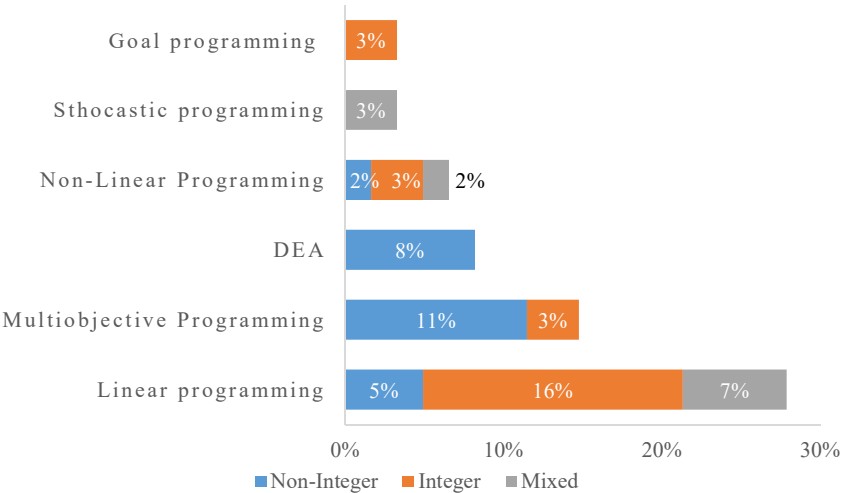

**Figure 7.** Most frequently used MODM methods used in the articles.

Primarily associated with AHP, mathematical models are also conventional approaches to select R&D projects. The 0–1 integer programming is the most frequently used one, appearing in 17 articles. Several other relevant papers use mathematical models. For instance, Wang and Hwang [17] formulate a fuzzy 0–1 integer programming model that can both manage uncertain and flexible parameters to define the optimal project portfolio. Bard et al. [44] employed 0–1 integer programming to evaluate both active and prospective R&D projects. They considered the full range of organizational, environmental, and technical concerns. Stummer and Heidenberger [32] describe a three-phase approach to support managers in reaching the most attractive project portfolio. First, it identifies worthy project proposals for further evaluation, keeping the number of projects within a manageable size for entering the subsequent phase. Second, a multiobjective integer linear programming model determines the solution space of all efficient portfolios. Third, it aims to find a portfolio that fits the decision-maker notions. Carlsson et al. [101] developed the trapezoidal fuzzy numbers estimation of future cash flows to value the alternatives on R&D projects. They presented a fuzzy mixed integer programming model and discussed how to use their methodology to support optimal R&D project selection in a corporate environment. Czajkowski and Jones [31] proposes a decision support modeling framework

for multi-project technology planning and project selection using 0–1 integer programming in which technical and explicitly assess the benefit interactions. Sun and Ma [41] developed and applied a heuristic packing-multiple boxes (PMB, or multi-knapsacks-model) model, based on several 0–1 integer programming methods, to pronounce both selecting and scheduling R&D projects.

**Table 4.** Main categories of MADM methods.

| Categories | Methods | Advantages | Disadvantages | Utilization |
|---|---|---|---|---|
| Multi-criteria value functions | Scoring, MAUT, CBA | Can incorporate preferences. The results are easy to understand. Some approaches are simple. | The preferences need to be precise. A lot of input is needed. | 13.2% |
| Outranking approaches | One novel model presented | It may take uncertainty and vagueness into account. Quantitative criteria may assume preference thresholds. | Do not weight the criteria in a systematic way. The outcome may be difficult do explain, since strengths and alternative are not directly identified. | 1.5% |
| Distance to ideal point | TOPSIS, VIKOR | Easy to use and program. The number and programming efforts of the steps remain the same regardless of the number of criteria and alternatives. | Its difficult to weight and keep consistency of judgement. Qualitative criteria are not easily handled. | 10.3% |
| Pairwise comparisons | AHP, ANP, DEMATEL | Not data intensive. Easy to use. Can easily handle with qualitative criteria. | Not recommended when there are several criteria and/or several alternatives, which should not be split into smaller comparison matrices. Depending on the number of comparisons, the process may be tiring and lead to inconsistency. | 35.3% |

Data Envelopment Analysis (DEA) appeared in 8% of the papers. Besides the work of Rabbani et al. [36], two more articles use DEA among their methods. Eilat et al. [33] developed an extended version of Data Envelopment Analysis (DEA) by integrating a balanced scorecard and the DEA itself for R&D project evaluation. Oral et al. [102] proposes a methodology for evaluating and selecting R&D projects. While the evaluation process relies on the DEA method, the selection process uses an ordinal scale throughout a method that uses model-based outranking.

A genetic algorithm is the main metaheuristic used to solve the mathematical models, appearing in 5% of the articles. Bhattacharyya et al. [30] presented a fuzzy multiobjective programming approach to aid the decision-makers in dealing with interdependences and uncertainty in R&D project selection. They presented a case study to demonstrate the proposed method where the solution is provided by a genetic algorithm (GA) and multiple objective genetic algorithms (MOGA). Eshlaghy and Razi [34] proposed a new approach of outranking relation in MCDA methods together with a data mining method for clustering and ranking the best R&D projects in a portfolio, presenting then a two-phase decision model for project portfolio selection problems. In the first phase, the clustering R&D projects in a portfolio with the most suitable projects specification combination uses a k-means algorithm. In the second phase, grey relational analysis (GRA) selects and evaluates the most efficient project in any cluster. Finally, a genetic algorithm (GA) calculates the Pareto front rank. Stewart [28] presented a solution for the project portfolio optimization problem using multiobjective programming and genetic algorithm.

For Malczewski [59], the classification of the methods should also follow the uncertainty related to the variables (see Figure 8). Thus, the decision-maker has to have a perfect comprehension of the decision environment for deterministic decision-making. If not, the probabilistic indication (lack of information) or fuzzy process (imprecision of semantic meanings). Interestingly, only deterministic approaches are the most frequently used ones since the R&D environment is turbulent and the results uncertain by nature.

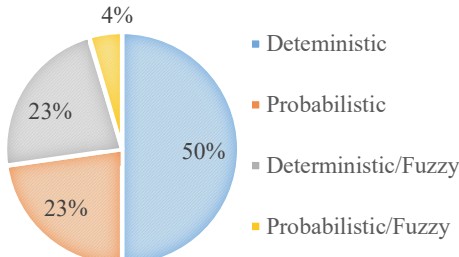

**Figure 8.** Uncertainty related to the variables.

Another feature of MCDM approaches is the full range of decision environments that have been employed over the last years [59]. Figure 9 highlights the main R&D PPS applications areas of the MCDM approaches.

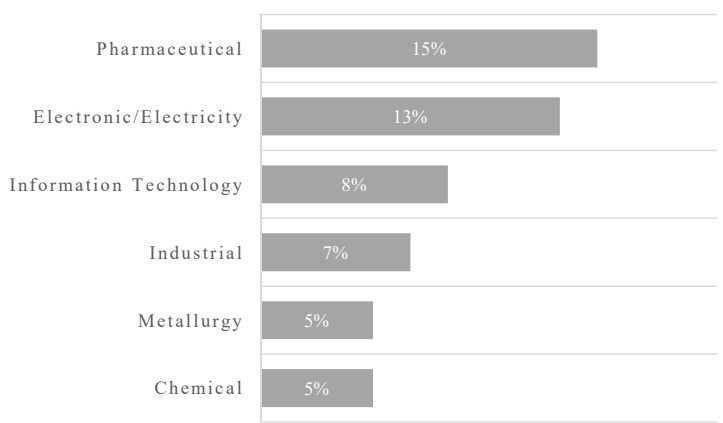

**Figure 9.** Main R&D application domains in MCDM-based R&D PPS.

Another interesting piece of information about the papers is the number of projects considered in the model. If the number of projects is excessively large, using pairwise methods, such as AHP, ANP, and DEMATEL, is not as good as to compare projects [12,15,103]. Another issue is that the largest the number of projects is, the higher the influence of uncertainty over the results will be. Thus fuzzy or stochastic approaches would be considered [9,26]. Figure 10 shows us the most common sizes of R&D project portfolios.

In Figure 11, we can observe that from 1970 to 1995, medium-sized portfolios accounted for 67% of all portfolios analyzed in the papers, against 22% of small-sized portfolios. However, from 1996 to the present small-sized portfolios has doubled its occurrence, representing 45% of all case studies addressed by the papers. This variation connects to the employment of MADM methods today, specially pairwise comparison methods, such as AHP and ANP. That also links to the more significant offer of software that facilitates the usage of MADM methods. Big-sized portfolios still represent fewer cases and may configure an opportunity to be explored by future papers, since it may represent the reality of many modern companies with big data-sets. In the case of big-sized portfolios, all papers use traditional MCDM approaches, except for Wei et al. [104], that used correlation analysis as an objective method.

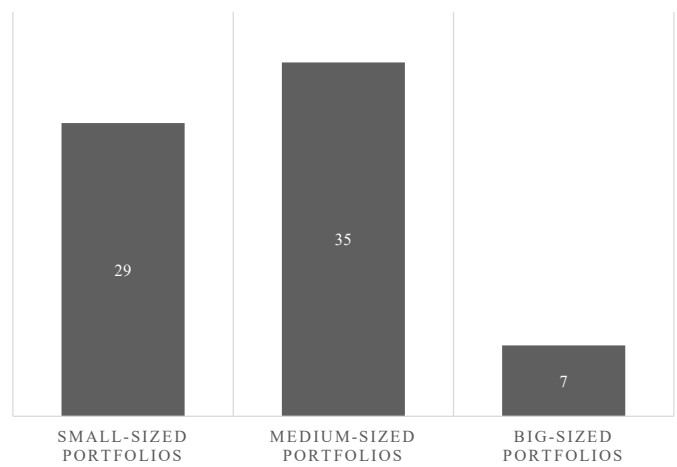

**Figure 10.** Number of projects in the portfolios.

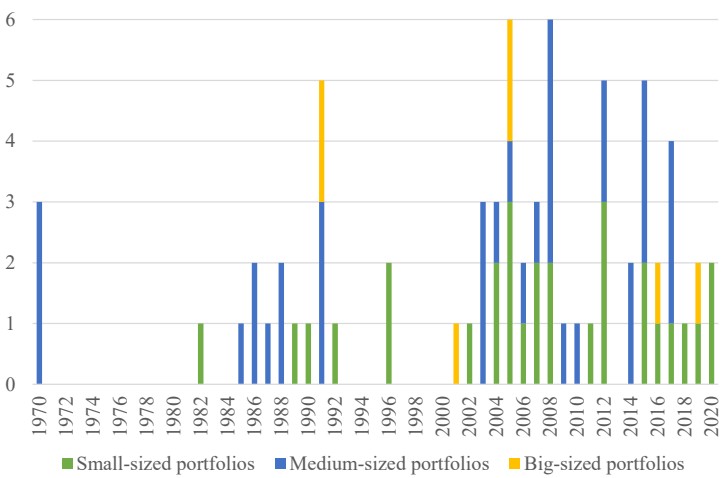

**Figure 11.** Yearly variation of the portfolio's size.

In some articles, researchers have used software, solvers, or programming languages to implement MCDM methods in R&D PPS. Figure 12 illustrates the most frequently used computational approaches to solve PPS problems. Complementary to the data of Figure 12, Excel is the most frequently used spreadsheet software (8% of all computational approaches), Lingo/Lindo is the most frequently used solver (16% of all computational approaches), followed by Cplex (3% of all computational approaches). In the case of dedicated software, Expert Choice is the most frequently used one (8 % of all computational approaches). Some old software did have continuous use in the past, such as Lotus 1-2-3 and Steuer's ADBase, both with three appearances each. Regarding programming languages, there are three no specified appearances, while Fortran, Pascal, and C++ appeared twice each one.

Notice that non-mathematical or easy-to-use models are not presented by the articles. In fact, this is a research opportunity for all PPS fields. According to Schiffels et al. [105], other companies seldom replicate quantitative approaches, and the same to black-box models in terms of acceptance by firms. Thus, managers frequently rely on simple decision rules, since easy-to-use approaches are not available. This opens a wide field of exploration, especially for small-profitable R&D companies, that cannot afford customized solutions.

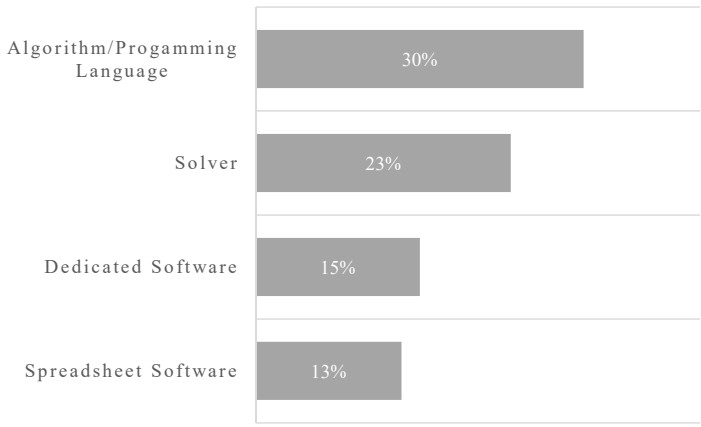

**Figure 12.** Programming languages, solvers and software used in R&D PPS.

Another relevant piece of information regards the criteria used by the approaches. Only 23 (35%) out of 65 articles explain the criteria used, while 42 articles (65%) do not explain the criteria used. Selecting and understanding the criteria used is a critical step in project portfolio selection and should not be avoided in real-world applications [49]. In fact, from all SLR articles, only Huang and Chu [69] and Souza et al. [51] propose methodologies for criteria selection in R&D PPS. In the first case, they present a Fuzzy-ANP for the Chinese government. In the second case, there is the integration between Fuzzy-AHP Extent Analysis and Fuzzy-based DEMATEL to find the best decision criteria in the case of the leading Brazilian R&D energy organization. This fact highlights potential research opportunities for future works and will be discussed with other opportunities later in this paper.

All classifications performed in this literature review are summarized by Figure 13 and we defined the methods and the applications in Table 5. Tables 6–8 present the articles that fall in each category.

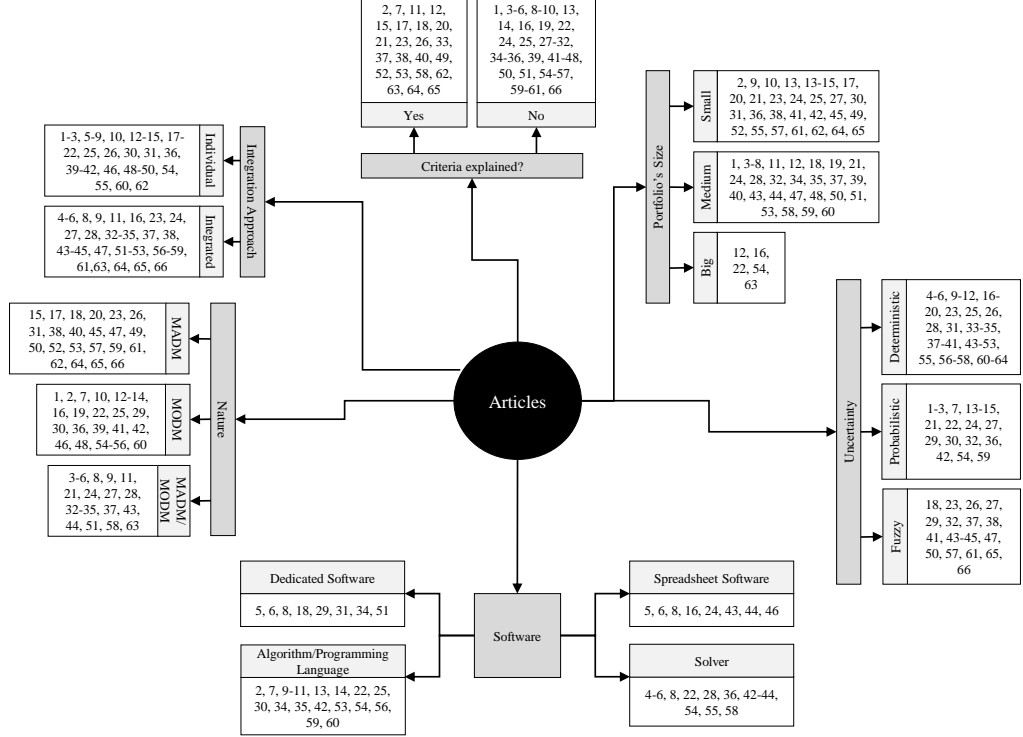

**Figure 13.** Map with part of the information about MCDM—based R&D PPS articles.

**Table 5.** Map description.

| Main Information | Description |
| --- | --- |
| Integration Approach | Individual when was used only one type of MCDM method, or Integrated when as used two or more methods. |
| Nature | We classify the methods into MADM, MODM, and integrated MADM/MODM approaches. MADM refers to methods that deal with a discrete and finite number of options designated by a predetermined set of criteria. MODM methods have no predetermined options, and the optimal option selection depends on an infinite and continuous number of circumstances subjected to a set of constraints. |
| Software | The classification points out the technology used to implement the R&D PPS methods: dedicated software, solvers, spreadsheet software, or algorithm/programming languages. |
| Uncertainty | The types of uncertainty were: Deterministic (when uncertainty is not considered), Probabilistic (when uncertainty is related to the lack of information), and Fuzzy (when their is imprecision of semantic meanings). Fuzzy numbers may be associated to Deterministic or, in rare cases, to Probabilistic approaches. |
| Portfolio sizes | We classified the portfolio sizes into Big (under 100 projects), Medium (between 9 and 99 projects), and Small (fewer than 8 projects). |
| Criteria explanation | We point out the papers that described the criteria used. |
| Application domain | Describe the decision environment where the R&D PPS is performed. |

**Table 6.** MCDM-based R&D PPS articles: number of projects, methods and application domains—Part 1/3.

| Author | Number of Projects | Methods | Application Domain |
| --- | --- | --- | --- |
| Bell and Read [106] | 40/12 and 22 | Linear Programming | Eletronic/Electricity and Chemical |
| Taylor et al. [42] | 7 | Non-Linear Integer Goal Programming | Textile |
| Madey and Dean [72] | 50 | MAUT, Mixed-Integer Non-Linear Programming, Multi-objective Programming, Preemptive Goal Programming | Aerospacial |
| Czajkowski and Jones [31] | 25 | DELPHI, Integer Linear Programming | Spacial |
| Liberatore [13] | 27 | AHP, CBA, Scoring, MAUT, Integer Linear Goal Programing | Chemical |
| Liberatore [15] | 27 | AHP, CBA, Integer Linear Programming | Chemical |
| Bard et al. [44] | 10 | Integer Linear Programming | Eletronic/Electricity |
| Liberatore [12] | 24 | AHP, CBA, Integer Linear Programming | Industrial |
| Ringuest and Graves [107] | 4 | DELPHI, Multi-objective Linear Programming, Goal Programming | Non-Specified |
| Ringuest and Graves [108] | 4 | Multi-objective Linear Programming | Non-Specified |
| Oral et al. [102] | 37 | DELPHI, Model-based Outranking Method, DEA | Metallurgy |
| Stewart [20] | 20, 50 and 150/250 | Non-linear Programming | Eletronic/Electricity |
| Graves and Ringuest [11] | 4 | Multi-objective Linear Programming | Non-Specified |
| Heidenberger [37] | 2 | Mixed-Integer Linear Programming | Non-Specified |
| Henig and Katz [109] | 5 | Not-specified MADM method | Biotechnology |
| Beaujon et al. [110] | 400 | Integer Linear Programming | Automotive |
| Meade and Presley [16] | 2 | ANP | Information Technology |
| Hsu et al. [73] | 12 | Fuzzy-AHP | Industrial |
| Stummer and Heidenberger [32] | 10 and 30 | Multi-objective Integer Linear Programming | Industrial |
| Kumar [74] | 6 | AHP | Research |
| Ringuest et al. [27] | 5 and 30 | Mean-Gini Analysis, Non-Linear Programming, Stochastic Dominance | Pharmaceutical |
| Gustafsson and Salo [19] | 1000 and 200 | Multi-objective Mixed-Integer Linear Programming | Non-Specified |
| Mohanty et al. [95] | 3 | Fuzzy-ANP; Fuzzy Cost Analysis | Metallurgy |
| Ringuest et al. [111] | 5 and 30 | Mean-Gini Analysis, Linear Programming | Pharmaceutical |
| Sun and Ma [41] | 8 | Integer Linear Programming | Non-Specified |
| Wang et al. [76] | Non-Specified | AHP, Fuzzy-Scoring | Information Technology |
| Karsak [25] | 6 | Fuzzy Integer Non-Linear Programming, ROA | Information Technology |
| Rabbani et al. [36] | 10 | AHP, Integer Linear Programming | Telecommunications |
| Carlsson et al. [101] | Non-Specified | Fuzzy Mixed-Integer Linear Programming | Non-Specified |

**Table 7.** MCDM-based R&D PPS articles: number of projects, methods and application domains—Part 2/3.

| Author | Number of Projects | Methods | Application Domain |
|---|---|---|---|
| Medaglia et al. [38] | 4 | Multi-objective Stochastic Linear Programming (Solved by Evolutionary Algorithm) | Non-Specified |
| Shin et al. [75] | 5 | AHP | Nuclear |
| Wang and Hwang [17] | 20 | Fuzzy Integer Linear Programming, Fuzzy-ROA (Compound Options) | Pharmaceutical |
| Bitman and Sharif [43] | Non-Specified | AHP, Scoring, BCG Matrix, BSC, DEA | Non-Specified |
| Conka et al. [10] | 14 | AHP, DEA, VTA | Non-Specified |
| Eilat et al. [33] | 60/50 | BSC, DEA, Linear Programming | Industrial |
| Fang et al. [112] | 3 | Mixed-Integer Stochastic Linear Programming | Non-Specified |
| Imoto et al. [40] | 18 | AHP, Fuzzy-Regression Analysis, PCA, Fuzzy Multi-objective Integer Linear Programming (Solved by Genetic Algorithm) | Metallurgy |
| Tolga et al. [22] | 6 | Fuzzy-AHP, Fuzzy-ROA | Eletronic/Electricity |
| Wu et al. [45] | 37 | Nash bargaining game | Non-Specified |
| Jung and Seo [29] | 14 | ANP | Government Sponsored |
| Bhattacharyya et al. [30] | 6 | Fuzzy Multi-objective Integer Linear Programming (Solved by Genetic Algorithm) | Civil, Mechanical, and others |
| Eckhause et al. [113] | 2 and 3 | Integer Linear Programming | Non-Specified |
| Hassanzadeh et al. [96] | 20 | Fuzzy Pay-off Method (ROA context), Fuzzy Integer Linear Programming | Pharmaceutical |
| Hassanzadeh et al. [97] | 20 | Fuzzy Pay-off Method (ROA context), Fuzzy Integer Linear Programming | Pharmaceutical |
| Mohaghar et al. [21] | 4 | Fuzzy-ANP, Fuzzy-TOPSIS | Manufacturing |
| Oral [35] | Non-Specified | E-DEA (Self-Efficiency DEA and Cross-Efficiency DEA) | Non-Specified |
| Collan and Luukka [23] | 20 | Fuzzy-TOPSIS, Fuzzy pay-off method (ROA context) | Pharmaceutical |
| Hassanzadeh et al. [24] | 14 | Multi-objective Integer Linear Programming | Information Technology |
| Bhattacharyya [114] | 5 | Grey Theory Sets | Civil, Mechanical and others |
| Collan et al. [77] | 20 | Fuzzy-TOPSIS | Pharmaceutical |
| Eshlaghy [34] | 20 | Grey Theory, Clustering Method (Solved by GA and K-Means) | Non-Specified |
| Jeng and Huang [79] | 5 | ANP, DELPHI, DEMATEL | Eletronic/Electricity |
| Karaveg et al. [78] | 45 | TOPSIS, SEM | Agriculture, Innovation, Textile and others |
| Arratia et al. [18] | 1500 | Mixed-integer Linear Programming | Private/Public Sector |
| Heydari T et al. [39] | 6 | Non-Linear Integer Goal Programming | Non-Specified |
| Stewart [28] | Non-Specified | Multi-objective Non-linear Programming (Solved by Reference Point, Genetic Algorithm, NIMBUS) | Non-Specified |

**Table 8.** MCDM-based R&D PPS articles: number of projects, methods and application domains—Part 3/3.

| Author | Number of Projects | Methods | Application Domain |
|---|---|---|---|
| Cheng et al. [46] | 5 | ANP, DEMATEL, COPRAS-G, Fuzzy Grey Relations | Eletronic/Electricity |
| Karasakal [103] | 60 | UTADIS (DEA based), AHP | Government Sponsored |
| Marcondes et al. [26] | 10 | Mean-Gini Analysis, Stochastic Dominance | Non-Specified |
| Montajabiha et al. [9] | 50 | Mixed-Integer Linear Programming | Pharmaceutical |
| Liang et al. [100] | 6 | TOPSIS, Pythagorean Fuzzy Theory | Private/Public Sector |
| Storch de Gracia et al. [115] | 5 | AHP | Energy |
| Wei et al. [104] | 100 | Correlation Analysis; Multi-objective Non-Linear Integer Programming | Military |
| Samanlioglu et al. [94] | 5 | Fuzzy-AHP; Fuzzy-VIKOR | Innovation |
| Aghdaie et al. [116] | 4 | SWARA; COPRAS | Electricity/Mechanical/Telecommunications/IT |
| Yalcin et al. [80] | Non-Specified | Fuzzy-TOPSIS; Fuzzy-DEMATEL | Non-Specified |

## 3. The Criteria Used in R&D PPS

In these 66 articles, authors 1 and 2 extracted the criteria through spreadsheet support and independent collection; after that, authors 3 and 4 revised and validated the theoretical list of criteria. We found 263 criteria employed in different R&D PPS contexts. The list is extensive, varying from commonly used criteria, such as net present value and value of return [36,44], to more different criteria, such as reputability of the project manager or the number of papers that could result from the research [10,74]. We listed all criteria references used, which are also available in the supplemental file.

A criterion expresses a perspective, and each perspective is instantiated through multiple criteria [43]. The selection of each criterion has to be precise in order to avoid duplication, overlapping, and misalignment with the organization's strategic goals. Thus, the selected criteria must be representative, significant, and indispensable for the project selection process [117]. In MCDM models, criteria may also appear as synonymous with attributes, objectives, or variables, which can be measured by quantitative/factual or qualitative/intangible data.

However, yet the 263 criteria are different in name and represented by distinct metrics, they express lesser scenario perspectives. Thus, in this paper, we summarize those 263 criteria in 23 criteria with broader scopes (or perspectives).

The grouping method is an adaptation of the one proposed by Jiro [118] and used in many grouping approaches [119,120]: the Affinity Diagram (also known as KJ Method, named after its author, Kawakita Jiro). By using the Affinity Diagram, we group criteria based on their natural relationships, obtained through brainstorming. Ten experts in R&D PPS acted in this process, being the steps described as follows:

- Step 1: The criteria split in 263 digital cards, and a document referencing and explaining each one.
- Step 2: Three experts received a third of the cards. Then the cards were placed in groups of affinity.
- Step 3: The experts then accessed the formed groups. They could move the cards among groups by arguing with the others.
- Step 4: A consensus occurred just when all three experts have stopped moving the cards.
- Step 5: 27 groups of criteria were obtained, named, and described by the experts by using references from the 263 initial criteria.
- Step 6: The next step was to verify this list. The verification intended to evaluate the internal consistency of the list of criteria. By consistency, we understand the lack of internal contradiction or intersection among the criteria. The verification step took into account the opinion of seven experts in R&D PPS. During the interview, they were asked to confront the actual list of criteria used by their organization with the 27 criteria proposed, and check if these new criteria could replace those already used. Their opinion consolidated within a single list of suggested of eventual modifications.
- Step 7: We formed a new list consisting of 23 criteria that all ten experts accepted all changes.

Mathematical and computational grouping/clustering approaches were not employed, such as K-Means Method and Hierarchical Clustering Algorithms. They would require great effort from the experts to perform quantitative or qualitative judgements to each criterion, due to the number of criteria and the non-existence of clustering attributes that would cover all criteria. It also would happen with clustering approaches based on graphs, such as spectral clustering. For instance, directly creating a symmetric adjacency matrix ($263 \times 263$) would require 34,453 comparison among criteria, which is not reasonable to be manually handled.

The new list of criteria is shown on Tables 9 and 10. The numbers of papers that used the criteria appear in the Utilization column. We present the profiles of the ten experts in Table 11.

**Table 9.** Theoretical list of criteria—1/2.

| # | Criteria | Description | Utilization |
|---|----------|-------------|-------------|
| 1 | Commercial and Market Risk (CMR) | It refers, in general, to the ambiguity of a project to generate commercial success [13,33,95]. | 3 (5%) |
| 2 | Competitiveness and Partnership (COP) | Measures a project's potential to improve the company's share in the market more than its competitors. It can be accomplished, for example, by using Science and Technology (S&T) policies or with the development, use, and commercialization of proprietary technology [73]. | 10 (16%) |
| 3 | Corporate Image (COI) | Describes the potential of a project to enhance the company's visibility to society or with a specific company or with an economic segment. Some authors, like Liberatore [13], used corporate image as a criterion. Other authors indirectly achieved this by pursuing other goals, such as the contribution of a project to the national economy [76]. | 5 (8%) |
| 4 | Environmental Impact (ENI) | Measures the capacity of a project to produce any environmental benefit [20,103]. Besides the internal context, it can also be associated with the external context, such as the project ecological relationships [43] or its sustainability [78]. | 5 (8%) |
| 5 | Extendibility (EXT) | It is related to a project's potential to improve the company's growth by adding new components or integrating the project with other public policies. It can be estimated, for illustration, by the applicability of a project results in other products and processes [16], the potential technical interaction with existing products [95] and the compatibility with other projects [13]. | 9 (15%) |
| 6 | External Environment Income (EEI) | It considers all factors and criteria that are not within the company and beyond its control, such as the existence of competitors [95], unexpected volatilities [9], and regulations [21,95]. | 28 (46%) |
| 7 | Financial Benefit (FIB) | It expresses the organization's financial return of the project and can be measured by different indicators, such as net present value (NPV) [36], the present value of return [44], real options value (ROV) [22] and others. | 41 (67%) |
| 8 | Financial Income (FII) | Relates to all financial resources required to implement the project, and they can be measured in terms of cost, budget, cash flow, total investment, and other metrics [12,25,30,46,108]. | 48 (79%) |
| 9 | Impact in Human Development (IHD) | Associates to any criteria correlated to the development and training of human resources [20,33]. | 9 (15%) |
| 10 | Internal Environment Income (IEI) | Includes the criteria connected with factors inside an organization, like workplace safety and manufacturing capability [16,46]. | 20 (33%) |
| 11 | Market Potential and Attractiveness (MPA) | Includes criteria exclusively related to the market and the receptivity by the market to the outcomes of the project [10,74], such as sales, market acceptance, interactions, trends, potential and possible market share [72,95]. | 3 (5%) |
| 12 | Material Resources (MAR) | Includes the criteria associated with resources that will be consumed, like raw material and energy [46,76]. | 38 (62%) |
| 13 | Non-Financial Benefit (NFB) | Expresses the non-financial gains of the project to an organizational, such as patents [29] and academic papers [10]. | 38 (62%) |
| 14 | Organizational Requirements (ORR) | Includes the criteria imposed by the organization, like the R&D objective, priority, congruence, and importance [33,40,41], clarity of definition [74] and, product life cycle [95]. | 17 (28%) |
| 15 | Quality Requirements (QTR) | It brings together all the criteria that can interfere with the overall quality of the project, such as customer feedback, customer satisfaction, and quality proposal [33,73], and expected utility [95]. | 21 (34%) |

As complement to the list of experts, a short description of all five organizations are given bellow:

- CNPq: The National Council for Scientific and Technological Development (CNPq) is a governmental agency belonging to the Brazilian Minister of Science, Technology, Innovations, and Communications (MCTIC). CNPq founded in 1951 with the main function of promoting scientific and technological research over the country. In 2015, CNPq invested US$ 623 million in R&D. From those, 87% was dedicated to research grants in Brazil and to Brazilians abroad.
- FINEP: The Financing Institution of Research and Innovation (FINEP) is a Brazilian public organization attached to the MCTIC. Created in 1967 to promote innovation

and R&D in Brazilian companies, universities, and public institutions. In 2018, FINEP invested US\$ 250 million in innovative initiatives. Several Brazilian agencies rely on FINEP resources to sponsor R&D projects, such as BNDES and CNPq.

- ANEEL: The Brazilian Electricity and Regulatory Agency (ANEEL) is an autarky established in 1997 below a special regime joined to the Brazilian Ministry of Mines and Energy. Its goal is to control the Brazilian electric sector. At the end of 2018, ANEEL approved a budget for energetic development in 2019 of US\$ 5.2 billion. From 2008 to 2017, ANEEL produced around US\$ 1.2 billion to finance R&D projects in the electricity sector and effectively utilized around 89%.

- BNDES: The Brazilian Development set (BNDES) started its activities in 1952, and today is one of the largest development sets in the world. It is the Brazilian federal government's largest instrument to finance long-term projects in all economic segments. Along with companies and public organizations, BNDES makes specific portfolio selections to promote innovation and national research and development. The last updated project portfolio selections available in its website add together resources up to US\$ 4 billion.

- ANP: The National Agency of Petroleum, Natural Gas, and Biofuels (ANP) was created in 1997 and is responsible for regulating the Brazilian activities in Petroleum, Natural Gas, and Biofuels. It is an autarky linked to the Brazilian Ministry of Mines and Energy. From 1998 to 2018, the agency has invested US\$ 4 billion in R&D projects. Only in 2018, the total investment sum up US\$ 147 million. Other organizations implement all ANP R&D investments in the segment. Petrobras, the 256th worldwide most innovative organization [4], was responsible for 74% of those investments in 2018.

**Table 10.** Theoretical list of criteria—2/2.

| # | Criteria | Description | Utilization |
|---|----------|-------------|-------------|
| 16 | Scope Risk (SCR) | It measures the project's results probability of staying outside its scope after the conclusion. Consequently, it can be associated with the risk of delay [34], additional costs [95] or unexpected interdependencies [114]. | 21 (34%) |
| 17 | Social Impact (SOI) | Measures the project's capacity to create social benefit [102,107]. It can also be associated with job creation opportunities [103], and the ethics or morality of the project [43]. | 7 (11%) |
| 18 | Strategic Fitness (STF) | Measures the project's capacity to meet the strategic goals of the company. It can be also described as strategic fit [101] and strategic need [95], for example. | 12 (20%) |
| 19 | Technical Contribution and Innovativeness | Indicates the project's potential to introduce new approaches to accomplish new technologies [79,102]. It can also be measured in terms of technological advancement [73], and creativity [76]. | 16 (26%) |
| 20 | Technical Issues and Constraints (TIC) | This criterion is related to the leading technologies used in the project and their impact or possible associated problems. So, the criteria can be exemplified as technological connections. [73], the technological difficulty [40] and type of technology [73]. | 9 (15%) |
| 21 | Technical Risk (TER) | It is generally related to the uncertainty associated with the technology or the probability of occurrence of technical problems [16,74]. | 4 (7%) |
| 22 | Timing Requirements (TIR) | It is associated with all criteria connecting with a time dimension, such as timing, project completion time, and time to market [13,16,39]. | 15 (25%) |
| 23 | Work Resources (WOR) | This criterion comprises the resources that will be utilized, such as labor and their necessary knowledge and experience [17,21] or employing a reputable leader or team [74]. | 40 (66%) |

**Table 11.** Experts, their qualifications and tasks.

| # | Experience in PPS in the Organization | Experience in PPS | Higher Education Degree | Participation as | Organization |
|---|---|---|---|---|---|
| E1 | 5 years | 5 years | M. Sc. | Project manager | ANP |
| E2 | 7 years | 8 years | M. Sc. | Project manager | ANP |
| E3 | 3 years | 3 years | Doctorate | Decision maker | CNPq |
| E4 | 9 years | 9 years | Doctorate | Decision maker | CNPq |
| E5 | 3 years | 10 years | Doctorate | Decision maker | FINEP |
| E6 | | | | Decision maker | FINEP |
| E7 | 17 years | 17 years | Doctorate | Decision maker | ANEEL |
| E8 | 3 years | 10 years | Doctorate | Decision maker | ANEEL |
| E9 | 3 years | 5 years | Doctorate | Board Director Member (Responsible for implementing guidelines for R&D PPS) | ANEEL |
| E10 | 3 years | 10 years | Doctorate | Project manager | BNDES |

## 4. Bibliometric Analysis

Based on the collected papers on MCDM methods employed in R&D PPS, in this section, we conducted a bibliometric analysis of the investigated papers. The data collected produce quantitative information about publications per year, the main authors, most productive countries, highly-cited papers, and leading journals on the topic. This bibliometric analysis intends to discover possible research trends and provide researchers and other practitioners a picture of the field. It is not intended to be a standard bibliometric analysis of the field, and we consider only the 66 previously discussed papers in our analysis.

Figure 14 compares the number of published articles on MCDM in R&D PPS over the years. We can graphically observe a warm-up in the field over 1982–1996 and a general increasing tendency in the number of published articles from 2000 to the present. We can also observe a statistically notable rise in the number of published papers from 1970 to 2019. It is expressed by the overall Person correlation coefficient (0.588 and $p = 0.000$, for $p < 0.01$). By analyzing the publication timeline in Figures 4, 5 and 11, the publication of the last three years, there is a tendency of concentration in MADM, integrated methodology approach, and small-size portfolio.

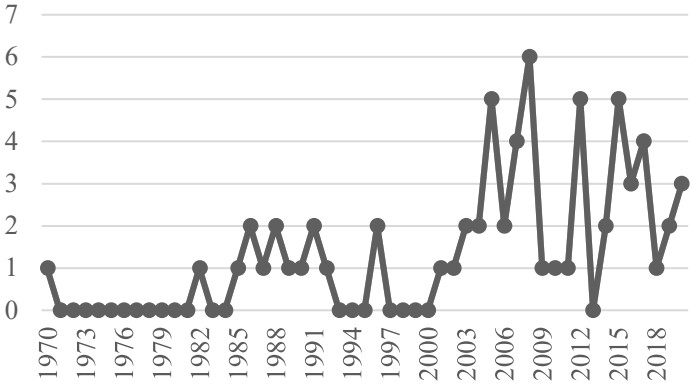

**Figure 14.** Publication pattern of MCDM applications in R&D PPS over the years 1970–2020.

Figure 15 shows the top-ten first authors in R&D PPS. The numbers are presented in terms of yearly citations average.

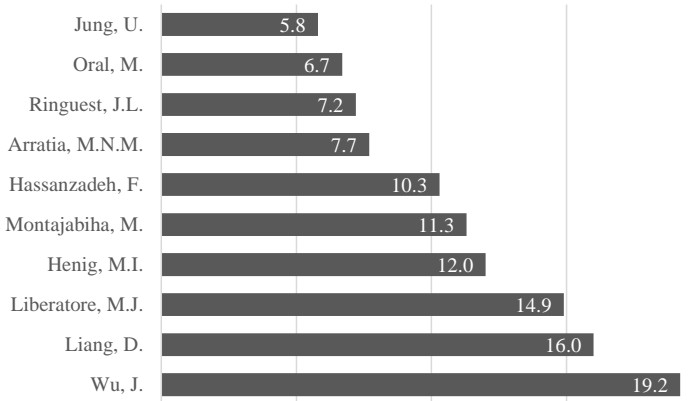

**Figure 15.** Most relevant correspondent authors on the subject considering their yearly citation average.

Additionally, over the same period, the authors publishing from the United States have been the most productive ones as they have contributed to 19 out of the 66 papers published in the period. Other countries, such as Turkey, Taiwan, China, Iran, India, and Finland, also contributed significantly. In Figure 16, we have displayed only countries with four contributions or more. Other countries contributed with 15 papers, which represents around 25% of all publications. Nevertheless, if we consider the country region, the most productive region was Asia with 47% of the studies followed by America with 34.8%, the other regions contribute 9.1% or less.

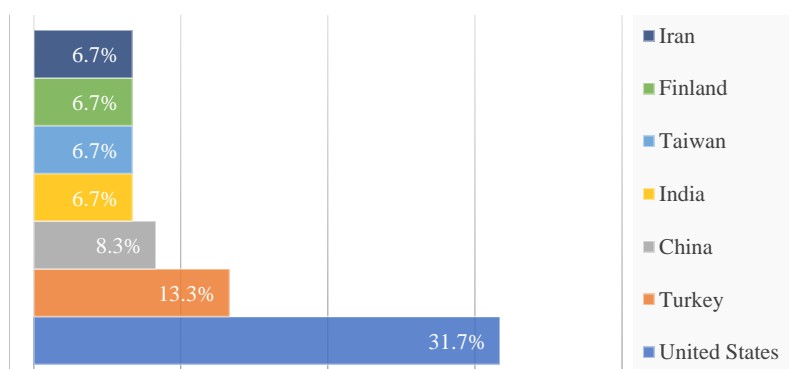

**Figure 16.** Top ten countries in MCDM-based R&D PPS.

In Table 12, the top ten papers are ordered by their total citations. Together, they sum 16% of all published articles, but contribute to 59% of all citations MCDM methods applied to R&D PPS. There is an indication of the number of citations per year, and the citation numbers collected from the database where the paper is available if the paper is available in both Web of Science and Scopus.

Observe that ranking articles based upon the total citation does not always match the average citation ranking. All top-cited papers appeared at least ten years ago. Typically, an influential paper establishes many citations after a while, such as Liang [100]. Over the review time frame, Figure 17 shows the leading journals on the R&D PPS MCDM applications. We highlight the first three journals. With 11 (17%) papers, the IEEE Transactions on Engineering Management contributed more, and the second was the European Journal of Operational Research with 5 (8%) papers, and the third was Omega with 3 (5%) papers.

**Table 12.** Most cited papers in MCDM-based R&D PPS from 1970 to 2019.

| Papers | Tota Citations | Citation per Year | Database |
|---|---|---|---|
| Meade and Presley [16] | 327 | 19.24 | WoS/Scopus |
| Liberatore [15] | 141 | 4.41 | WoS/Scopus |
| Wang and Hwang [17] | 136 | 11.33 | WoS/Scopus |
| Eilat et al. [33] | 132 | 12.00 | WoS/Scopus |
| Oral et al. [102] | 131 | 4.68 | WoS |
| Stummer and Heidenberger [32] | 123 | 7.69 | WoS/Scopus |
| Mohanty et al. [95] | 122 | 8.71 | WoS |
| Carlsoon et al. [101] | 102 | 8.50 | WoS/Scopus |
| Bard et al. [44] | 73 | 2.35 | Scopus |
| Medaglia et al. [38] | 69 | 5.75 | WoS |

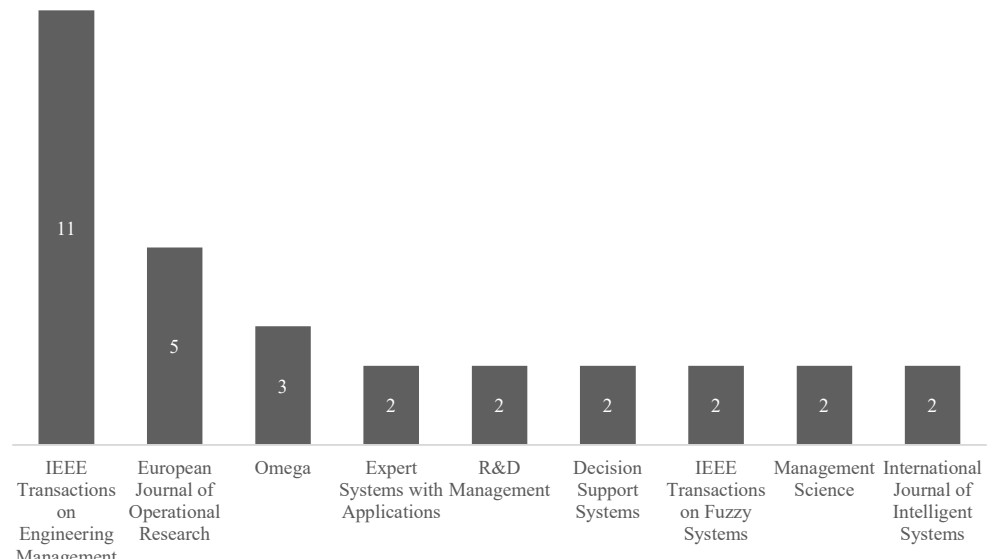

**Figure 17.** Most representative journals in MCDM-based R&D PPS.

## 5. Opportunities and New Paths

We also present opportunities that could be explored by researchers and practitioners of R&D PPS in their future works. Formulating research question is an appropriate way to highlight and guide future research, while preventing researchers from pursuing unnecessary and no longer used directions [121]. Thus, the formulation of clear research questions, derived from data and insights obtained from the papers in the literature review, is a guide for future work. The research questions divisions appear in two groups: research questions presented by recent articles (last three years) as opportunities for future works (last three years), and research questions that could not be answered by the articles considered in the SLR (see, Table 13). In the case of this last type of questions, all of them were prior discussed in their correspondent topics.

**Table 13.** Research questions to guide further research.

| **Research Questions That Could Not Be Answered by The articles Considered in the SLR** |
| --- |

| 1 | How historical data from previously executed project portfolios could be used to better select new projects? None of the presented models incorporate data-driven decision making approaches to their frameworks. Nonetheless, several classification and regression approaches, such as Random Forest and Support Vector Machines/Regressors could be used to enhance the selected portfolio. |
| 2 | Which MCDM approaches are more suitable to select project portfolios from several project proposals? Large portfolios are representative in many countries, especially developing countries, where R&D investments highly depend on public and governmental agencies through calls, which normally receive several project proposals. The R&D PPS field does not seem to be already impacted by big data. |
| 3 | Which are the best MCDM approaches to small-profitable R&D companies? The MCDM approaches proposed in the literature are, sometimes, far from the reality of many companies, that do not have the personnel to use them, nor sufficient money to provide software running those approaches [105]. |
| 4 | How to define, a priori, criteria for selecting projects? Only two articles from the performed SLR present methodologies for criteria selection [51,69]. However, criteria selection is one of the most important steps on PPS in general [49]. |
| 5 | How to employ outranking MADM methods (such as PROMETHEE and ELECTRE) in the R&D PPS context? There is no result in literature for these methods when applied to R&D PPS in general. However, uncertainty, vagueness and preference thresholds are still characteristics of several R&D project portfolios. |
| 6 | How to select the best method for each R&D PPS application? Several articles do not explain why they have chosen some methods to PPS among all possible options. When this is the case, they mainly use usage frequency, which is the case of standalone AHP applications [74,75,104]. A framework that helps researchers to select the method to use is a lacuna not only in R&D PPS. It seems to be an opportunity in the hole MCDM context [50]. |

| **Research Questions Presented by Recent Articles as Opportunities for Future Works** |
| --- |

| 1 | Is the three-way decisions-based ideal solution, proposed by Liang et al. [100], suitable to other fuzzy environments, rather than the new Pythagorean fuzzy environment? |
| 2 | Cheng et al. [46] suggested integrating DEMATEL and CFPR-ANP to weight the criteria and then COPRAS-G to rank the projects in an electronic company. Is this approach suitable to organizations of other segments? |
| 3 | How does the method proposed by Karasakal and Aker [103] respond to different data sets and different reference sets? They propose integrating Interval AHP, DEA and UTADIS to select governmental R&D projects. |
| 4 | How to consider resource constraints, interrelations, or mutual-exclusion to select projects? This approach appeared in Marcondes et al. [26] that suggest a using Mean-Gini and stochastic dominance. A further question, how the approach responds to a real portfolio? |
| 5 | How would the multi-objective mixed-integer linear programming, proposed by Arratia et al. [18], respond to other features? Such as: uncertainty, resource-allocation in planning-horizon, scheduling and risk-assessment mechanisms |

## 6. Conclusions

Many companies rely on R&D in order to have chances of standing in the market. However, since the resources are limited, only a few project proposals will integrate the company's project portfolio. In this context, MCDM methods have risen as essential tools to help decision-makers select R&D projects. Hence, in this paper, we attended several objectives whose summary of findings we present in Table 14.

This paper has reviewed the existing research literature in MCDM-based R&D PPS in order to guide the community of academics and contribute to knowledge accumulation and creation concerning the usage of MCDM methods in R&D PPS. It also is a good starter reading for those that are taking the first steps in the topic. For this purpose, we summarized all approaches to MCDM used on the papers in R&D PPS from 1970 to 2020. We classified them according to their methods, nature, integration approach, and uncertainty related to the variables. There is a categorization of portfolios according to the application domain and size. Opportunities and possibilities of future works were discussed, and bibliometric analysis of the papers was also performed.

**Table 14.** Summary of findings.

| Research Questions | Findings |
|---|---|
| RQ1: Methods. | We presented the nature of the alternatives in Figure 2. The methods used are displayed in Figure 3. Figures 6 and 7 showed the most frequently used MADM (Multi-attribute decision-making) and MODM (Multi-objective decision-making) methods. Tables 6–8 presented the usage of those methods. Finally, we displayed how the researchers used those methods in time in Figure 5 and Figure 8, the uncertainty related to the variables. |
| RQ2: Portfolios. | Medium-sized portfolios stand for 67% of all portfolios analyzed in the papers, small-sized portfolios 22%, and big-sized portfolios 11%. The pharmaceutical application is the most explored domain, as shown in Figure 9. The programming language used by researchers is shown in Figure 12. The researchers defined the criteria in 34.8% of the papers. |
| RQ3: Research Field. | Table 12 showed the most cited articles, Figure 15 displayed the most relevant correspondent author, and Figure 16 showed the Unites States as the leading country; however, Asia had the most regional affiliation. |
| RQ4: Whole data. | The correlations analysis showed that articles using MADM approaches tend to explain better the criteria used, While MODM focuses more on the model itself and leaves this kind of detail (explaining the criteria, for example) unattended. We highlighted this by the moderate negative correlation ($-0.51$) between linear programming (So far, the most frequently used MODM method) and the presence of explained criteria on the paper. Some MCDM methods also seem to be primarily used as supporting methods, which is the case of CBA, which presents a $+0.86$ correlation coefficient with the usage of Integrated Approaches. Similarly, DEMATEL is used as an auxiliary to ANP. |
| RQ5: Criteria used. | We presented the theoretical list of criteria comprised of 23 criteria in Tables 9 and 10. |
| RQ6: Opportunities and trends. | We explored the opportunities and new paths in subsection 5, and also we presented research questions to guide further research in Table 13. |

From 1970 to 2020, MADM and MODM methods emerged in similar proportions. However, MADM methods tend to be more used nowadays than in the past. Among MADM methods, AHP (23%) is the most employed one, followed by ANP (9%), ROA (9%), TOPSIS (8%), and DELPHI methods (6%). Linear integer programming (28%) is the most applied MODM approach. According to the uncertainty related to the variables, deterministic decision making is preferred over probabilistic and fuzzy approaches, appearing in 49% of all papers. Pharmaceutical and Electronic Electricity R&D domains are the most investigated ones, and each one is studied in 13% of the papers. Programming Language (21%) is the most commonly used way to solve R&D PPS problems, and the most studied portfolios (52%) are medium-sized portfolios.

Fuzzy sets are used in 27% of the papers. Despite the popularity of triangular fuzzy numbers among MCDM practitioners, in R&D PPS other membership functions and fuzzy environments are preferred. However, all fuzzy set applications are used as a way to tackle imprecise information, mainly originated from the use of verbal scales when using MADM approaches, and to facilitate the aggregation of experts judgments.

Likewise, from the SLR, we have collected 263 criteria used by the 66 articles about MCDM-based R&D PPS, published from 1970 to 2020. Those criteria were condensed into a shorter list of criteria, each one representing a different perspective. The whole process was conducted with the assistance of experts from five Brazilian R&D organizations that together manage portfolios valued around US$ 5 billion a year, which accounts for 38% of all Brazilian annual expenditures in R&D projects.

From the bibliometric analysis, we have also obtained relevant results. Responsible for 41% of all papers, the United States is the country with the most significant quantity of contributions to MCDM-based R&D PPS. However, Asia is the most active region in R&D PPS, holding 45% of all papers, followed by Americas with 35%. The top ten most-cited papers account for 59% of all citations in the area, and IEEE Transactions on Engineering Management is the journal publishing most of the papers (18%).

Limitations still need to be overcome in future works. For example, further research should consider scientific databases other than the Web of Science and Scopus, and although the extent Boolean combination was high, we may have missed some papers or important

conference papers of R&D PPS. Snowball techniques are also suitable, as they may enlarge the number of papers used in the SLR/BA. The literature review we presented also did not investigate any methods other than MCDM methods, such as data-driven and logic-based approaches. Those also seem viable possibilities for future expansions of the scope of the work we presented.

**Supplementary Materials:** The following materials are available in the supplementary data, as bellow: Availability of data, code and other materials click File S1. Prisma 2020 checklist click Table S1. Prisma 2020 for Abstracts Checklist click Table S2.

**Author Contributions:** In this research, all authors contribute in some way. D.G.B.d.S. and E.A.d.S. screened each record and were responsible for almost all investigation, writing, and original draft preparation. N.Y.S. and C.E.S.d.S. did the conceptualization, validation, and supervision. All authors have read and agreed to the published version of the manuscript.

**Funding:** This work was supported in part by the National Council for Scientific and Technological Development (CNPq) of Brazil, and in part by the Research Supporting Foundation of Minas Gerais State (FAPEMIG) of Brazil, and in part by the São Paulo Reaserch Foundation (FAPESP) of Brazil.

**Institutional Review Board Statement:** Not applicable.

**Informed Consent Statement:** Not applicable.

**Data Availability Statement:** The data is contained within the article and addressed in Supplementary Materials.

**Acknowledgments:** Dedication: this paper is in loving memory of Fábio Carneiro Mokarzel (1941–2021).

**Conflicts of Interest:** All authors have approved the manuscript and agree with its submission. Furthermore, we confirm that this manuscript has not been previously published and is not considered for publication in any other journal. Therefore, no conflict of interest needs to be disclosed.

## Abbreviations

The following abbreviations are used in this manuscript:

| | |
|---|---|
| AHP | Analytic Hierarchy Process |
| ANEEL | Brazilian Electricity and Regulatory Agency (in Portuguese: Agência Nacional de Energia Elétrica) |
| ANP | Analytic Network Process Or National Agency of Petroleum, Natural Gas and Bio-fuels (in Portuguese: Agência Nacional do Petróleo, Gás Natural e Biocombustíveis) |
| ARA | Additive Ration Assessment |
| BCG | Boston Consulting Group |
| BNDES | Brazilian Development Bank (in Portuguese: Banco Nacional de Desenvolvimento Econômico e Social) |
| BSC | Balanced Scorecard |
| CBA | Cost-Benefit Analysis |
| CBR | Case Based Reasoning |
| CFPR | Consistent Fuzzy Preference Relations |
| CNPq | National Council for Scientific and Technological Development (in Portuguese: Conselho Nacional de Desenvolvimento Científico e Tecnológico) |
| COPRAS | Complex Proportional Assessment |
| CP | Compromise Programming |
| DEA | Data Envelopment Analysis |
| DEMATEL | Decision-Making Trial and Evaluation Laboratory |
| ELECTRE | ELimination and Choice Expressing Reality (in French: ELimination Et Choix Traduisant la REalité) |

| | |
|---|---|
| EVAMIX | Evaluation of Mixed Data |
| FINEP | Financing Institution of Research and Innovation (in Portuguese: Financiadora de Inovação e Pesquisa) |
| GA | Genetic Algorithm |
| GRA | Gray Relational Analysis |
| LINMAP | Linear Programming Technique for Multidimensional Analysis and Preference |
| MADA | Multi-Attribute Decision Analysis or Multi-Attribute Decision Aiding |
| MADM | Multi-Attribute Decision Making |
| MAUT | Multi-Attribute Utility Theory |
| MAVT | Multi-Attribute Value Theory |
| MCDA | Multi-Criteria Decision Analysis or Multi-Criteria Decision Aiding |
| MCDM | Multi-Criteria Decision Making |
| MCTIC | Brazilian Minister of Science, Technology,Innovations and Communication (in Portuguese: Ministério da Ciência, Tecnologia, Inovações e Comunicações) |
| MME | Brazilian Ministry of Mines and Energy (in Portuguese: Ministério de Minas e Energia) |
| MODA | Multi-Objective Decision Analysis or Multi-Objective Decision Aiding |
| MODM | Multi-Objective Decision Making |
| MOGA | Multiple objective genetic algorithm |
| MOORA | Multi-Objective Optimization on the basis of Ration Analysis |
| MULTIMOORA | Multiplicative form with Multi-Objective Optimization on the basis of Ration Analysis |
| NAIADE | Novel Approach to Imprecise Assessment and Decision Environments |
| PMBOK | Project Management Body of Knowledge |
| PPM | Project Portfolio Management |
| PPS | Project Portfolio Selection |
| PROMETHEE | Preference Ranking Organization Method for Enrichment of Evaluations |
| REMBRANDT | Ratio Estimation in Magnitudes or Decibels to Rate Alternatives which are Non-Dominated |
| R&D | Research and Development |
| ROA | Real Options Analysis |
| SAW | Simple Additive Weighting |
| SLR | Systematic Literature Review |
| SMART | Simple Multi-Attribute Rating Technique |
| SWARA | Step-wise Weight Assessment Ration Analysis |
| TOPSIS | Technique for Order Preference by Similarity to Ideal Solution |
| UIS | UNESCO Institute for Statistics |
| UTADIS | Utilities Additives Discriminates |
| VIKOR | Multi-criteria Optimization and Compromise Solution (in Serbian: VIseKriterijumska Optimizacija I Kompro-misno Resenj) |
| WASPAS | Weighted Aggregated Sum Product Assessment |
| WPM | Weighted Product Method |
| WSM | Weighted Sum Method |

## Appendix A

Articles included in the literature review, Tables A1 and A2.

**Table A1.** Articles included in the literature review—Part 1/2.

| Author | Title | Year |
| --- | --- | --- |
| Bell and Read [106] | The application of a research project selection method | 1970 |
| Taylor et al. [42] | R and D Project Selection and Manpower Allocation with Integer Non-Linear Goal Programming | 1982 |
| Madey and Dean [72] | Strategic Planning for Investment in R&D usiong decision analysis and mathematical programming | 1985 |
| Czajkowski and Jones [31] | Selecting Interrelated R&D projects in Space Technology Planning | 1986 |
| Liberatore [13] | R&D project selection | 1986 |
| Liberatore [15] | Extension of the Analytic Hierarchy Process for Industrial R&D Project Selection and Resource Allocation | 1987 |
| Bard et al. [44] | An Interactive Approach to R&D Project Selection and Termination | 1988 |
| Liberatore [12] | An expert support system for R&D project selection | 1988 |
| Ringuest and Graves [107] | The Linear Multi-Objective R&D Project Selection Problem | 1989 |
| Ringuest and Graves [108] | The Linear R&D Project Selection Problem: An Alternative to Net Present Value | 1990 |
| Oral et al. [102] | A Methodology for Collective Evaluation and Selection of Industrial Research and Development projects | 1991 |
| Stewart [20] | A multi-criteria decision support system for r&d project selection | 1991 |
| Graves and Ringuest [11] | Choosing the best solution in an R&D project selection problem with multiple objectives | 1992 |
| Heidenberger [37] | Dynamic project selection and funding under risk: A decision tree based MILP approach | 1996 |
| Henig and Katz [109] | R&D project selection: A decision process approach | 1996 |
| Beaujon et al. [110] | Balancing and optimizing a portfolio of R&D projects | 2001 |
| Meade and Presley [16] | R&D project selection using the analytic network process | 2002 |
| Hsu et al. [73] | Fuzzy multiple criteria selection of government-sponsored frontier technology R&D projects | 2003 |
| Stummer and Heidenberger [32] | Interactive R&D portfolio analysis with project interdependencies and time profiles of multiple objectives | 2003 |
| Kumar [74] | AHP-based formal system for R&D project evaluation | 2004 |
| Ringuest et al. [27] | Mean-Gini analysis in R&D portfolio selection | 2004 |
| Gustafsson and Salo [19] | Contingent portfolio programming for the management of risky projects | 2005 |
| Mohanty et al. [95] | A fuzzy ANP-based approach to R&D project selection: a case study | 2005 |
| Ringuest et al. [111] | Formulating optimal R&D portfolios | 2005 |
| Sun and Ma [41] | A packing-multiple-boxes model for R&D project selection and scheduling | 2005 |
| Wang et al. [76] | Analytic hierarchy process with fuzzy scoring in evaluating multidisciplinary R&D projects in China | 2005 |
| Karsak [25] | A generalized fuzzy optimization framework for R&D project selection using real options valuation | 2006 |
| Rabbani et al. [36] | A comprehensive model for R and D project portfolio selection with zero-one linear goal-programming | 2006 |
| Carlsson et al. [101] | A fuzzy approach to R&D project portfolio selection | 2007 |
| Medaglia et al. [38] | A multiobjective evolutionary approach for linearly constrained project selection under uncertainty | 2007 |
| Shin et al. [75] | Applying the analytic hierarchy process to evaluation of the national nuclear R&D projects: The case of Korea | 2007 |
| Wang and Hwang [17] | A fuzzy set approach for R&D portfolio selection using a real options valuation model | 2007 |
| Bitman and Sharif [43] | A conceptual framework for ranking R&D projects | 2008 |
| Conka et al. [10] | A combined decision model for R&D project portfolio selection | 2008 |
| Eilat et al. [33] | R&D project evaluation: An integrated DEA and balanced scorecard approach | 2008 |
| Fang et al. [112] | A mixed R&D projects and securities portfolio selection model | 2008 |
| Imoto et al. [40] | Fuzzy regression model of R&D project evaluation | 2008 |
| Tolga et al. [22] | Fuzzy multiattribute evaluation of R&D projects using a real options valuation model | 2008 |
| Wu et al. [45] | Bargaining game model in the evaluation of decision making units | 2009 |
| Jung and Seo [29] | An ANP approach for R&D project evaluation based on interdependencies between research objectives and evaluation criteria | 2010 |
| Bhattacharyya et al. [30] | Fuzzy R&D portfolio selection of interdependent projects | 2011 |
| Eckhause et al. [113] | An Integer Programming Approach for Evaluating R&D Funding Decisions With Optimal Budget Allocations | 2012 |
| Hassanzadeh et al. [96] | A Practical Approach to R&D Portfolio Selection Using the Fuzzy Pay-Off Method | 2012 |
| Hassanzadeh et al. [97] | A practical R&D selection model using fuzzy pay-off method | 2012 |

**Table A2.** Articles included in the literature review—Part 2/2.

| Author | Title | Year |
|---|---|---|
| Mohaghar et al. [21] | An integrated approach of Fuzzy ANP and Fuzzy TOPSIS for R&D project selection: A case study | 2012 |
| Oral [35] | Action research contextualizes DEA in a multi-organizational decision-making process | 2012 |
| Collan and Luukka [23] | Evaluating R&D Projects as Investments by Using an Overall Ranking From Four New Fuzzy Similarity Measure-Based TOPSIS Variants | 2014 |
| Hassanzadeh et al. [24] | Robust optimization for interactive multiobjective programming with imprecise information applied to R&D project portfolio selection | 2014 |
| Bhattacharyya [114] | A Grey Theory Based Multiple Attribute Approach for R&D Project Portfolio Selection | 2015 |
| Collan et al. [77] | New Closeness Coefficients for Fuzzy Similarity Based Fuzzy TOPSIS: An Approach Combining Fuzzy Entropy and Multidistance | 2015 |
| Eshlaghy [34] | A hybrid grey-based k-means and genetic algorithm for project selection | 2015 |
| Jeng and Huang [79] | Strategic project portfolio selection for national research institutes | 2015 |
| Karaveg et al. [78] | A combined technique using SEM and TOPSIS for the commercialization capability of R&D project evaluation | 2015 |
| Arratia et al. [18] | Static R&D project portfolio selection in public organizations | 2016 |
| Heydari T et al. [39] | Developing and solving an one-zero non-linear goal programming model to R and D portfolio project selection with interactions between projects | 2016 |
| Stewart [28] | Multiple objective project portfolio selection based on reference points | 2016 |
| Cheng et al. [46] | A Consistent Fuzzy Preference Relations Based ANP Model for R&D Project Selection | 2017 |
| Karasakal [103] | A multicriteria sorting approach based on data envelopment analysis for R&D project selection problem | 2017 |
| Marcondes et al. [26] | Using mean-Gini and stochastic dominance to choose project portfolios with parameter uncertainty | 2017 |
| Montajabiha et al. [9] | A robust algorithm for project portfolio selection problem using real options valuation | 2017 |
| Liang et al. [100] | Method for three-way decisions using ideal TOPSIS solutions at Pythagorean fuzzy information | 2018 |
| Storch de Gracia et al. [115] | Multicriteria methodology and hierarchical innovation in the energy sector: The Project Management Institute approach | 2019 |
| Wei et al. [104] | Model and Data-Driven System Portfolio Selection | 2019 |
| Samanlioglu et al. [94] | An intelligent approach for the evaluation of innovation projects | 2020 |
| Aghdaie et al. [116] | Decision making on exigent issues in organizations: a case study on R&D projects | 2020 |
| Yalcin et al. [80] | Research and Development Project Selection via IF-DEMATEL and IF-TOPSIS | 2020 |

**Appendix B**

Figure A1 shows the PRISMA 2020 flow diagram for new systematic reviews which included searches of databases and registers only.

**PRISMA 2020 flow diagram for new systematic reviews which included searches of databases and registers only**

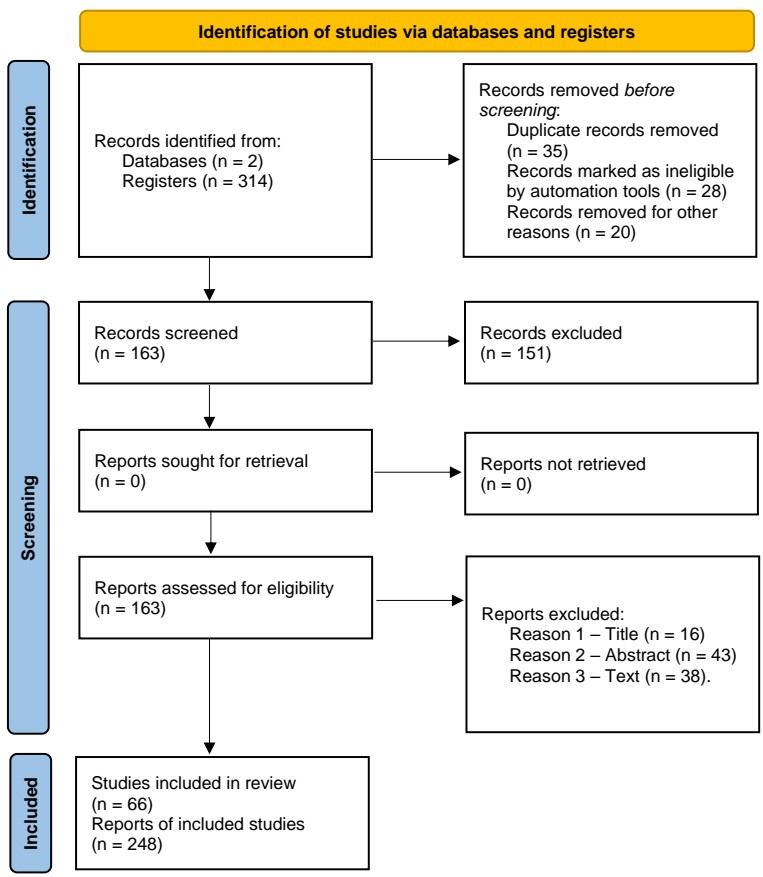

**Figure A1.** Prisma diagram.

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
