# Peer review of "MCDM-Based R&D Project Selection: A Systematic Literature Review"

_sustainability, doi:10.3390/su132111626_

Round 1

Reviewer 1 Report

The paper belongs to an interesting line of research. I put below some suggestions for potential improvements.

  1. This paper was need to specify the innovation in the manuscript. I suggest rebuild and extend introduction section. The Authors’ approach and practical case study must be positioned in the light of other up to date studies.
  2. The methods stated very clearly.
  3. Please explain the summarization of your research findings more clearly.
  4. The conclusions are a bit limiting. I suggest the authors to expand the conclusions or suggestions a bit.
  5. Please check some mistakes Line 115, 122 and 126 ==> “In Section ??”

Reviewer 2 Report

I had the chance to study this review article and it is quite interesting. However, I have few concerns before accepting it.

  1. It's a review article but above the title, they mentioned that it's an article. Please correct it.
  2. Many typos in abbreviations. See MCDM, it already defined in the abstract, yet the author used full form in line 82. Please correct it.
  3. What is PRISMA in line 132? Please explain? the authors explained it few words later.
  4. Make corrections in line 126.
  5. Line 140, check the sentence structure.
  6. Line 167, 168. Check the sentence structure.
  7. The authors raised some questions at the start of the paper? Did they succeed in finding the answers? if yes, they should discuss in the conclusion.
  8. The authors took papes only from two sources, does it points towards any weakness?
  9. There are many generalizations of fuzzy sets, will including them in the study have an impact or not? Please discuss in conclusion.
  10. The similarity is 27% excluding references. Please reduce it.

Reviewer 3 Report

Please find attached the Review Report.

Round 2

Reviewer 2 Report

The paper is well revised and should be accepted.

Reviewer 3 Report

Dear author(s),

The revised version of the manuscript entitled “MCDM based R&D project selection: A systematic literature review” (Manuscript ID: sustainability-1391242) improved in a suitable manner. The author(s) considered most of the suggested changes as expressed throughout the first review round. As well, the author(s) provided appropriate replies for each of the concern. Therefore, the paper deserves to be published in current form.